# UrbanMLLM: Joint Learning of Cross-view Imagery for Urban Understanding

Xin Zhang [1] [*]    Tianjian Ouyang Xu [1] [*]    Yu Shang [1] [*]    Qingmin Liao [1]    Yong Li [1] [†]

## Abstract

Comprehensive urban understanding requires integrating macroscopic spatial structure with fine-grained street-level semantics. However, existing urban Multimodal Large Language Models (MLLMs) primarily rely on satellite imagery, limiting their ability to capture detailed urban appearance and cross-view relationships. We propose **UrbanMLLM**, a unified MLLM that jointly learns from satellite and street-view imagery for cross-view urban perception and reasoning. To support this, we construct a large-scale dataset with paired cross-view urban images, geospatial alignment, and textual annotations. UrbanMLLM introduces a cross-view perceiver to explicitly model interactions between satellite and street-view representations, and adopts a structural interleaved pre-training paradigm that organizes cross-view image–text content as coherent urban documents to enhance cross-view knowledge fusion. We evaluate UrbanMLLM on 13 diverse urban understanding tasks spanning satellite, street-view, and cross-view settings. Experimental results demonstrate consistent improvements over strong open-source and proprietary MLLMs, highlighting effectiveness and scalability of UrbanMLLM for urban environment understanding.

## 1. Introduction

Urban imagery has been widely used to understand cities in terms of urban spatial structure, functionality, and socio-economic status. The advancement in computer vision and multimodal learning has driven the utilization of multimodal urban data for urban understanding tasks, such as scene classification (Kuckreja et al., 2024; Mall et al., 2024; Zhang et al., 2024b), scene geo-localization (Vivanco Cepeda et al., 2024; Xu et al., 2024; Feng et al., 2025a), urban indicator prediction (Fan et al., 2023; Hao et al., 2025; Liu et al., 2025a;b), etc. More recently, benefiting from the impressive performance and generalizability of large language models, multimodal large language models (MLLMs) have shown great potential for effectively solving diverse multimodal tasks in a "one-for-all" manner.

In urban studies, existing MLLMs predominantly focus on remote sensing tasks (Kuckreja et al., 2024; Luo et al., 2024; Bazi et al., 2024; Muhtar et al., 2024). While remote sensing imagery provides a macroscopic overview of urban layouts, it lacks detailed contextual information about urban elements, thereby limiting these MLLMs to high-level tasks like land use classification. In contrast, street-view imagery captures fine-grained appearance and complementary details such as building facades and heights. However, no existing MLLMs in urban areas have explored integrating street-view imagery to enhance urban understanding. To achieve comprehensive urban comprehension, it is crucial to jointly learn from both large-scale satellite imagery and detailed ground-level street-view imagery within a unified model.

Achieving this goal necessitates addressing two primary challenges. First, there is a dearth of well-organized multimodal urban data, as existing public datasets lack cross-view imagery paired with corresponding textual annotations, precluding joint learning for MLLMs. Second, the joint learning paradigm itself poses a challenge: traditional MLLM frameworks (Liu et al., 2024c; Chen et al., 2024) encode visual features from different views separately, aligning them only with their respective texts. This approach fails to capture the rich complementary relationships between satellite and street-view imagery, leaving their information isolated.

In this work, we address the above two challenges and introduce a novel urban MLLM jointly learning from satellite and street-view imagery and associated textual data. Given that existing publicly available datasets about urban imagery (Luo et al., 2024; Astruc et al., 2024; Han et al., 2024; Xi et al., 2023) commonly lack paired cross-view images and large-scale annotated text information, we first collect a large-scale multimodal urban imagery dataset. Our dataset covers the whole United States, comprising paired-up satellite-view and street-view images, together with geotags and annotated textual descriptions. We propose two key designs to break the isolation of visual knowledge and facil-

---

[*]Equal contribution [1]Tsinghua University. Correspondence to: Yong Li <liyong07@tsinghua.edu.cn>.

*Proceedings of the 43rd International Conference on Machine Learning*, Seoul, South Korea. PMLR 306, 2026. Copyright 2026 by the author(s).

*Table 1.* Comparison of UrbanMLLM with existing works for urban understanding in terms of data sources and targeted tasks.

| Method Type | Model | Data | | Task | | |
|---|---|---|---|---|---|---|
| | | Satellite Image | Street View Image | Perception | Reasoning | Prediction |
| CLIP-based | RemoteCLIP | ✓ | ✗ | ✗ | ✗ | ✓ |
| | UrbanCLIP | ✓ | ✗ | ✗ | ✗ | ✓ |
| | UrbanVL | ✓ | ✓ | ✗ | ✗ | ✓ |
| MLLM-based | GeoChat | ✓ | ✗ | ✓ | ✓ | ✗ |
| | LHRS-Bot | ✓ | ✗ | ✓ | ✓ | ✗ |
| | SkysenseGPT | ✓ | ✗ | ✓ | ✓ | ✗ |
| | UrbanMLLM | ✓ | ✓ | ✓ | ✓ | ✓ |

itate the mutual learning of cross-view urban imagery. The first one is about the model architecture, where we propose a cross-view perceiver module that bridges the paired-up satellite and street-view visual features through a cross-attention mechanism. This design *explicitly* facilitates the exchange of information between the region-level context of satellite imagery and the fine-grained appearance details of street-view imagery. For example, the injection of street-view information into the satellite feature can provide more detailed urban region contexts. The second part is a novel interleaved pre-training paradigm to enhance the mutual learning between cross-view imagery. In detail, we design coherent image-text documents that interleave the satellite image with matched street-view images and associated textual descriptions, forming a comprehensive profile of an urban region. Such an interleaved training corpus helps MLLMs *implicitly* learn the relationship between different-view urban imagery via in-context learning. Through the explicit and implicit mutual learning between cross-view urban imagery, our proposed **UrbanMLLM** is expected to overcome the visual isolation issue, benefiting the comprehensive understanding of urban environments.

To comprehensively evaluate MLLMs' urban understanding capabilities, we established a benchmark encompassing 13 diverse tasks: urban perception (scene classification, geo-localization), reasoning (object, spatial relationship, landmark), and prediction (indicator prediction), based on single-view or cross-view urban imagery. Extensive experiments on this benchmark validate UrbanMLLM's noticeable superiority across a wide range of urban understanding tasks. This work serves as a foundational technique for addressing diverse urban-related challenges requiring comprehensive visual understanding.

Our main contributions can be summarized as follows:

- We propose a brand MLLM architecture with a designed cross-perceiver module to facilitate cross-fusion of the complementary visual context from satellite-view and street-view imagery.
- We construct a novel interleaved pre-training corpus that links satellite and street-view imagery through geo-

location relationships, and propose a training paradigm that implicitly promotes mutual learning between cross-view imagery.

- We establish a comprehensive benchmark comprising 13 diverse urban understanding tasks spanning single-view and cross-view urban imagery. Extensive experiments demonstrate our model achieves substantial improvements in urban understanding over both open-source and closed-source MLLMs. Code is available at `https://github.com/tsinghua-fib-lab/UrbanMLLM`.

## 2. Related Work

### 2.1. Multimodal Large Language Models

Leveraging the success of LLMs, Multimodal Large Language Models integrate visual and textual understanding for complex reasoning. MLLMs broadly divide into closed-source, like GPT-4o (Achiam et al., 2023), Gemini (Reid, 2024), and Qwen-VL (Bai et al., 2023), which benefit from vast proprietary corpora for general multimodal comprehension. Open-source MLLMs, typically smaller and extending architectures like LLaVA (Liu et al., 2024c), advance through improved architectures (e.g., dual vision encoders (Li et al., 2024b), sophisticated visual adapters (Cha et al., 2024), MoE (Li et al., 2024a)) or better pre-training data (e.g., interleaved (Lin et al., 2024)). In contrast, our work introduces a novel MLLM architecture with a cross-view perceiver module for enhanced cross-view information fusion and contributes a unique interleaved pre-training corpus for urban environments.

### 2.2. Multimodal Models For Urban Understanding

Understanding the urban environment usually requires multimodal information from diverse sources, such as satellite-view images, street-view images, POI information and geo-locations, etc. Existing methods in urban study can be categorized into two types: CLIP-based methods and MLLM-based methods, as shown in Table 1. From the data aspect, existing methods based urban imagery in urban studies all focus on satellite images while overlooking using the street-

view imagery for urban understanding. CLIP-based methods are mostly developed based on the contrastive learning strategy used in CLIP (Radford et al., 2021), such as training with satellite image-text pairs (Liu et al., 2024a; Yan et al., 2024), street-view image-text pairs (Hao et al., 2025) and satellite-view and street-view image pairs (Mall et al., 2024; Ouyang et al., 2024). These works can only deal with prediction tasks such as indicator prediction via end-to-end fine-tuning, but fail to conduct perception and reasoning tasks. Another line of research focuses on developing specialized MLLMs for problem-solving in the urban domain. Existing models, such as GeoChat (Kuckreja et al., 2024), SkysenseGPT (Luo et al., 2024), H$^2$RSVLM (Pang et al., 2024), and EarthGPT (Zhang et al., 2024a) only leverage remote sensing data including satellite images and annotated text for model learning. These models are capable of handling remote sensing perception and reasoning tasks but fail to deal with prediction tasks such as indicator predictions. As for other urban computing tasks, there are indeed some specialized models, such as Sample4Geo (Deuser et al., 2023) and MFRGN (Wang et al., 2024b). But these models are designed for specific tasks and can not be generalized to other urban understanding tasks. Furthermore, relying solely on region-level knowledge is insufficient to capture the complexities of urban environments, thereby limiting their applications for a wide range of urban understanding tasks. In contrast, our work proposes a novel learning paradigm based on cross-view urban image-text data which is capable of solving both remote sensing and street-view tasks.

## 3. Methodology

### 3.1. Overview

Traditional MLLMs, trained with separate image-text data, suffer from knowledge isolation between images, limiting their holistic urban environment comprehension. To overcome this, we first construct a dedicated interleaved dataset for pre-training and instruction dataset through a human–LLM collaborative pipeline. Building upon this foundation, we propose two key designs to facilitate a comprehensive understanding of cross-view urban imagery. Firstly, we introduce a cross-view perceiver module into the MLLM architecture, explicitly enabling satellite and street-view visual contexts to complement each other. Secondly, we propose a novel interleaved pre-training paradigm that leverages structurally interleaved urban image-text contexts, integrating cross-view imagery with textual descriptions to enhance joint learning.

### 3.2. Data Construction

To enhance model's capacity to address complex and diverse urban challenges, we propose a streamlined and systematic pipeline for urban data construction through human–AI collaboration. As shown in Figure 1, pipeline consists of four sequential stages: data collection, data annotation, data construction, and data cleaning.

Our data collection involved compiling over 2 million street view images via Google Maps API (Google, 2024) and nationwide satellite imagery from ESRI. This was complemented by socioeconomic data from diverse sources including WorldPop, NIH, and U.S. government entities. Additionally, we adapted public datasets such as Visual Genome (Krishna et al., 2016) and Place Pulse (Dubey et al., 2016) to build a comprehensive urban dataset. The data annotation stage involves LLM-assisted semantic labeling and refinement of spatially aligned data into image-text pairs. Subsequently, data construction employs a hybrid approach of rule-based templates and LLM-generated question-answering to create multimodal instructional data. For data cleaning, a rigorous human-AI hybrid validation process is implemented for image caption and instruction data quality. This process includes initial consistency assessment by VILA-1.5-40B and LLaVA-Next-34B, subsequent evaluation of flagged cases by GPT-4o, and ultimate human verification by a graduate-level annotator for persistent issues, with problematic captions being regenerated. In the 10K sample subset, the CLIP score shows only a marginal increase from 29.99535 to 29.99571, reflecting a 0.0012% improvement. This suggests that erroneous captions have minimal impact on the overall dataset. Additionally, the two-stage MLLM verification identifies just 1.3 mismatched image-caption pairs per thousand, further confirming the truthfulness of the generated captions.

Through this comprehensive process, we have constructed a large-scale dataset: an interleaved image-text dataset comprising 4.5 million matched pairs of satellite and street view images, where each satellite image is paired with three to six corresponding street view images. Besides, we have developed an instruction dataset containing approximately 700,000 question-answer pairs for SFT. These datasets collectively cover a wide range of urban understanding tasks, including perception, reasoning, and numerical prediction, with detailed task descriptions outlined below. As shown in the Figure 6, we provide an overview of the diverse satellite, street-view, and cross-view tasks in our instruct-tuning dataset, spanning SC, OR, LR, SRR, GL, and IP.

- **Satellite Imagery Tasks (SI)** includes Scene Classification (SC), Object Reasoning (OR), Spatial Relationship Reasoning (SRR), Geo-Localization (GL), Indicator Prediction (IP), population density prediction (Pop) and nightlight intensity prediction (Nightlight) are the sub-task of Indicator Prediction. Single Scene Classification (Single) and Multi-Scene Classification (Multi) are the sub-tasks of Scene Classification.

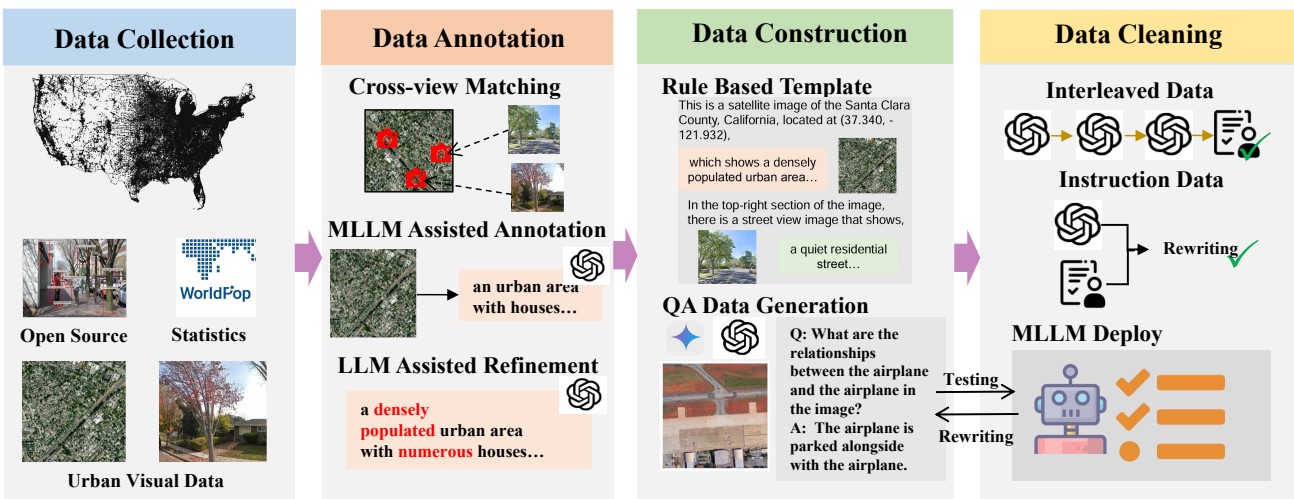

*Figure 1.* The pipeline for constructing the dataset, encompassing four key stages: data collection, data annotation, data construction, and data cleaning.

- **Street View Imagery Tasks (SVI)** includes Scene Classification (SC), Object Reasoning (OR), Landmark Recognition (LR), Spatial Relationship Reasoning (SRR), Geo-Localization (GL), Indicator Prediction (IP). Predicting the beautiful (BF) and wealthy (WE) level are the subtasks of Indicator Prediction.
- **Cross-View Tasks (CV)** includes Spatial Relationship Reasoning (SRR) and Indicator Prediction (IP). Predicting the median income (Med. income), poverty ratio (Pov. ratio), and total population (Population) levels are the sub-tasks of Indicator Prediction.

### 3.3. Cross-view Perceiver-enhanced UrbanMLLM

Current MLLMs in urban studies primarily focus on remote sensing tasks. These models are typically developed by directly fine-tuning general-purpose MLLMs (e.g., LLaVA) on satellite image-text pairs. However, effective urban understanding not only requires comprehending region-level knowledge from satellite imagery but also detailed contexts from street-view imagery. Unfortunately, this objective is hard to achieve with the classical MLLM architecture, where images are individually encoded, failing to receive visual knowledge from relevant images.

Aiming to address the visual knowledge isolation issue, we introduce a *cross-view perceiver* module $g_\zeta(\cdot)$ to promote the awareness of urban imagery from other views during the visual encoding stage. The architecture of the cross-view perceiver is illustrated in the left part of Figure 2. This module performs four key operations: (1) fusing the street-view image embedding into the satellite image embedding (denoted as "SVI to SI attn."); (2) fusing the satellite image embedding into the street-view image embedding (denoted as "SI to SVI attn."); (3) applying a gating mechanism to

adaptively combine the original and fused features; and (4) using an MLP to align the visual feature space with the text feature space of the central LLM.

When both satellite and street-view images exist in the multimodal input, let $I_{si}$ denote a satellite image and $\{I_{svi}^i\}_{i=1}^n$ represent $n$ paired street-view images. The forward process consists of the following steps:

*Step 1:* Both types of images are initially processed by a pre-trained visual encoder $f_\phi(\cdot)$, resulting in visual features $\boldsymbol{f}_{si}$ and $\{\boldsymbol{f}_{svi}^i\}_{i=1}^n$, respectively.

*Step 2:* For each street-view image $I_{svi}^i$, we inject the paired satellite image features into the street-view image feature through a cross-attention operation, obtaining the fused feature $\boldsymbol{e}_{si\to svi}^i$. For the satellite image, we first apply an average-pooling on the visual features of $n$ paired street-view images: $\widetilde{\boldsymbol{f}}_{svi} = \text{Pooling}(\{\boldsymbol{f}_{svi}^i\}_{i=1}^n)$. Then we fuse the pooled street-view feature $\widetilde{\boldsymbol{f}}_{svi}$ with the satellite feature $\boldsymbol{f}_{si}$, obtaining the fused feature $\boldsymbol{e}_{svi\to si}$.

*Step 3:* In each branch, the fused visual feature is adaptively combined with the original feature using a gating mechanism implemented with a one-layer MLP.

*Step 4:* The final visual embeddings $\mathbf{V}_{svi}^i$ and $\mathbf{V}_{si}$ are then obtained after a visual adapter (two-layer MLP).

The whole operation of the cross-view perceiver is formu-

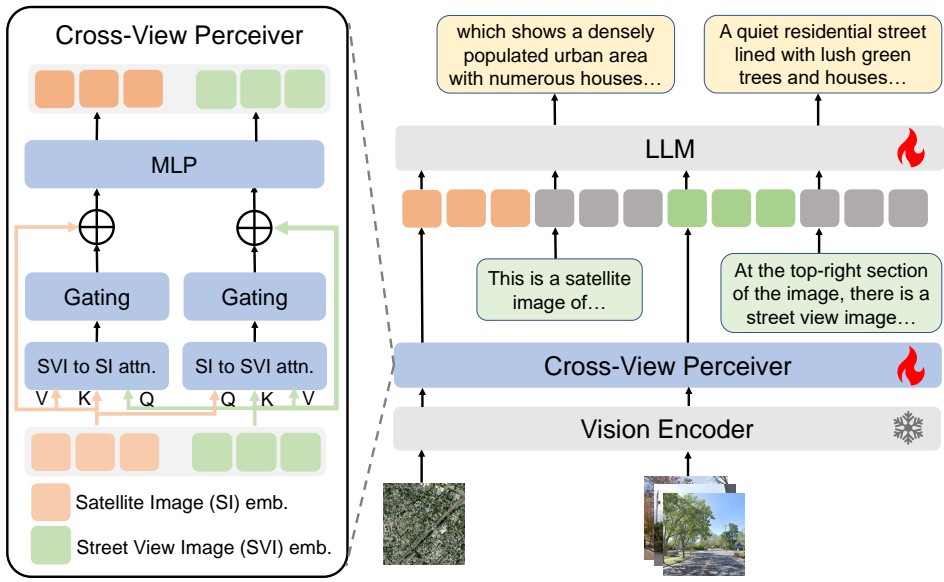

*Figure 2.* The architecture of UrbanMLLM introducing the cross-view perceiver to learn cross-view visual representations.

lated as follows:

$$e^i_{si \to svi} = \text{Linear}(\text{Softmax}(\frac{\boldsymbol{f}_{si}\boldsymbol{f}^i_{svi}}{\sqrt{d_k}})\boldsymbol{f}^i_{svi}), \quad (1)$$

$$\mathbf{V}^i_{svi} = \text{MLP}(\text{Gating}(e^i_{si \to svi}) + \boldsymbol{f}^i_{svi}), \quad (2)$$

$$e_{svi \to si} = \text{Linear}(\text{Softmax}(\frac{\widetilde{\boldsymbol{f}}_{svi}\boldsymbol{f}_{si}}{\sqrt{d_k}})\boldsymbol{f}_{si}), \quad (3)$$

$$\mathbf{V}_{si} = \text{MLP}(\text{Gating}(e_{svi \to si}) + \boldsymbol{f}_{si}). \quad (4)$$

In this way, the visual embedding fed to the central LLM is enhanced by the visual context from another view of the same urban region, which possesses a more comprehensive urban context. If only single-view imagery is present in the multimodal input, the cross-view perceiver module uses duplicate single-view images as input to ensure compatibility with the model's architecture, even in the absence of cross-view information. Although cross-attention mechanisms are prevalent, our cross-view perceiver marks their novel application specifically for unifying satellite and street-view imagery comprehension.

### 3.4. Interleaved Urban Context-based Pre-training

Existing MLLM research shows that pre-training on interleaved data, compared to traditional image-text pairs (Lin et al., 2024), significantly enhances performance by fostering semantic connections and contextual relationships across multiple images. This advantage directly aligns with our goal of jointly learning from cross-view urban imagery for a comprehensive understanding. For example, supplementing macroscopic satellite imagery with detailed street-view information is crucial for tasks like accurate geo-location prediction, where region-level visual information

alone is insufficient.

Motivated by this, we introduce an urban context-based interleaved training paradigm tailored for urban understanding tasks. The core of this design is the construction of multimodal interleaved urban imagery-text documents as the training corpus. We first collect a large-scale satellite and street-view imagery individually across the United States and perform cross-view matching based on geotags (e.g., located county, longitude and latitude), creating a paired cross-view urban imagery set $\mathcal{S} = \{(I_{si}, I^1_{svi}, I^2_{svi}, ..., I^n_{svi})|n \in \mathbb{Z}^+\}$. We then employ an advanced open-source MLLM InternVL (Chen et al., 2024) with carefully crafted prompts to efficiently generate textual descriptions for each image. To further enhance the text quality, we have also taken a human-AI collaboration pipeline to refine the annotated texts. Specifically, we first use two other powerful open-source MLLMs, VILA-1.5-40B and LLaVA-Next-34B to judge if the annotated text matches with the image. If either of them thinks it is not a match, we send the image to GPT-4o to re-generate the textual descriptions.

Next, we link the cross-view images based on their geographical relationships and integrate their corresponding textual descriptions and geotags, forming a comprehensive urban profile for each element in $\mathcal{S}$. Training on such interleaved multimodal urban data benefits the MLLM in fusing the relational knowledge between cross-view imagery into visual features. Assuming that the interleaved document contains $K$ ordered urban images $\boldsymbol{I} = \{I_k\}^K_{k=1}$ interleaved with a $T$-length word sequence $\mathbf{w} = \{w_t\}^T_{t=1}$ tokenized by a $\theta$-parameterized LLM. The $k$-th image is successively processed by a frozen visual encoder $f_\phi(\cdot)$ and the cross-view

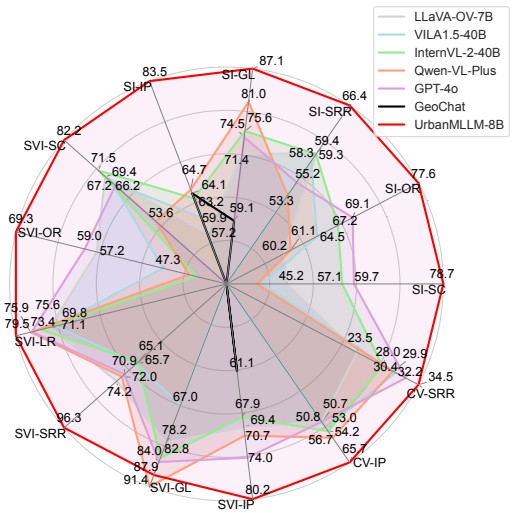

*Figure 3.* UrbanMLLM consistently outperforms existing open-sourced and closed-sourced MLLMs. SI, SVI and CV stand for satellite-view tasks, street-view tasks, and cross-view tasks.

perceiver $g_\zeta(\cdot)$ into $L$-length image tokens $\mathbf{V}_k = \{\mathbf{v}_l\}_{l=1}^L$. Denote $K(t)$ as the image index before the $t$-th word token. The pre-training objective of UrbanMLLM is to predict the next word token with preceding image and word tokens:

$$\mathcal{L}(\Theta = \{\theta, \zeta\}, \mathbf{w}, \boldsymbol{I}) = -\mathbb{E}_t[\log p_\Theta(\mathbf{w}_t | \mathbf{w}_{<t}, \mathbf{V}_{<K(t)})], \tag{5}$$

$$\mathbf{V}_{K(t)} = g_\zeta \circ f_\phi(I_{K(t)}). \tag{6}$$

## 4. Experiments

In this section, we conduct experiments to answer the following research questions:

- **RQ1:** How does UrbanMLLM perform compared with existing baseline models on satellite-view and street-view urban imagery understanding tasks?

- **RQ2:** Can UrbanMLLM accurately perform cross-view urban understanding tasks, such as socioeconomic indicator prediction and spatial relationship reasoning?

- **RQ3:** How effective are the key architectural designs of UrbanMLLM, including the cross-view perceiver and the interleaved pre-training strategy?

- **RQ4:** Does UrbanMLLM generalize well to external benchmarks without additional training?

### 4.1. Experimental Setup

**Implementation.** We initialize our model's weights using the pretrained VILA-1.5. The training process consists of two stages: in the first stage, we train on the curated interleaved pretraining dataset with a batch size of 8 for one

epoch. For the second stage, we fine-tune the model on the curated SFT dataset for one epoch. The image resolutions of satellite and street view images are both $512 \times 512$, while satellite images have a zoom level of 15. About 4.5 million images are used in the pre-training stage and 1 million images are used in the SFT stage. The baseline implementations are presented in the appendix.

**Metrics** The satellite multi-scene classification task is evaluated using the F1-score. For the indicator prediction task, accuracy with a margin of 2 is employed. All other tasks are assessed via QA accuracy with a fixed correct answer.

### 4.2. Results

We compare the performance of our proposed UrbanMLLM with baselines on three types of tasks: satellite imagery understanding, street view imagery understanding and cross-view understanding task on Table 2, 3, 4 and Figure 3. Based on these results, we have the following observations:

- **UrbanMLLM achieves the best performance across both satellite view and street view understanding tasks.** UrbanMLLM significantly outperforms optimal baselines without fine-tuning on satellite-view tasks, demonstrating 24.8% improvement in perception (e.g., scene classification, geo-localization) and 13.8% improvement in fine-grained object/spatial reasoning. For more challenging indicator prediction, our model surpasses supervised CLIP-based models by 12.1%. On street-view tasks, UrbanMLLM shows 15.5% improvement in fine-grained object/spatial reasoning and 0.5% improvement in indicator prediction compared to optimal baselines. To ensure a more rigorous comparison, we also fine-tuned the VILA1.5-8B model on our curated dataset. As demonstrated by our results, our proposed model consistently outperforms the fine-tuned VILA1.5-8b across a variety of urban image understanding tasks. This superior performance is attributed to our key designs including the interleaved text-image pre-training strategy and the novel cross-view feature fusion module designed for cross-view urban imagery learning.

- **UrbanMLLM achieves overall superior performance in cross-view understanding tasks.** As presented in Table 4, UrbanMLLM consistently outperforms baseline models across most cross-view tasks, particularly excelling in challenging tasks like poverty ratio prediction, spatial relationship reasoning, median income prediction, and population prediction. Furthermore, UrbanMLLM achieves an average 3.3% improvement over GPT-4o on four cross-view imagery tasks, demonstrating a very competitive performance. However, on poverty prediction tasks, GPT-4o has higher performance, which may be

*Table 2.* Performance comparisons on satellite imagery-based urban understanding tasks. "Failed" denotes cases where the model failed to produce valid numerical predictions under the required evaluation format.

| Satellite Imagery Task | SC | | OR | SRR | GL | IP | |
| --- | --- | --- | --- | --- | --- | --- | --- |
| Subtasks | Single | Multi | | | | Pop | Nightlight |
| RemoteCLIP | - | - | - | - | - | 0.766 | 0.688 |
| OpenAICLIP | - | - | - | - | - | 0.782 | 0.708 |
| VILA1.5-8B | 0.629 | 0.157 | 0.619 | 0.380 | 0.589 | 0.710 | 0.455 |
| QwenVL-2.5-7B | 0.521 | 0.199 | 0.576 | 0.329 | 0.502 | 0.555 | 0.300 |
| Qwen3-VL-32B | 0.529 | 0.314 | 0.401 | 0.582 | 0.458 | 0.617 | 0.302 |
| LLaVA-N-34B | 0.574 | 0.220 | 0.629 | 0.588 | 0.608 | 0.597 | Failed |
| InternVL-2-40B | 0.664 | 0.479 | 0.672 | 0.593 | 0.756 | 0.632 | Failed |
| Qwen-VL-Plus | 0.589 | 0.191 | 0.611 | 0.533 | 0.810 | 0.647 | Failed |
| GPT-4o | 0.680 | 0.513 | 0.691 | 0.552 | 0.745 | 0.484 | Failed |
| Gemini-3.1-Pro | 0.649 | 0.565 | 0.725 | 0.591 | 0.867 | 0.451 | 0.562 |
| GeoChat | 0.435 | 0.214 | 0.528 | 0.404 | 0.591 | 0.641 | Failed |
| LHRS-Bot | 0.439 | 0.128 | 0.568 | 0.386 | 0.243 | 0.533 | 0.449 |
| VILA1.5-8B (fine-tuned) | 0.779 | 0.759 | 0.797 | **0.686** | 0.846 | 0.855 | 0.787 |
| UrbanMLLM-3B | 0.766 | 0.729 | 0.784 | 0.584 | 0.829 | 0.847 | 0.744 |
| UrbanMLLM-8B | **0.787** | **0.776** | **0.800** | 0.664 | **0.871** | **0.872** | **0.798** |

*Table 3.* Performance comparisons on street view imagery-based urban understanding tasks.

| Street View Task | SC | OR | LR | SRR | GL | IP | |
| --- | --- | --- | --- | --- | --- | --- | --- |
| Subtasks | | | | | | BF | WE |
| RemoteCLIP | - | - | - | - | - | 0.767 | 0.725 |
| OpenAICLIP | - | - | - | - | - | 0.788 | 0.561 |
| VILA1.5-8B | 0.483 | 0.398 | 0.701 | 0.654 | 0.685 | 0.309 | 0.460 |
| QwenVL-2.5-7B | 0.512 | 0.587 | 0.699 | 0.605 | 0.722 | 0.677 | 0.647 |
| Qwen3-VL-32B | 0.499 | 0.633 | 0.723 | 0.725 | 0.652 | 0.819 | 0.705 |
| LLaVA-N-34B | **0.870** | 0.548 | 0.691 | 0.775 | 0.637 | 0.757 | 0.727 |
| InternVL-2-40B | 0.715 | 0.423 | 0.734 | 0.651 | 0.828 | 0.662 | 0.747 |
| Qwen-VL-Plus | 0.536 | 0.434 | 0.759 | 0.720 | **0.914** | 0.635 | 0.724 |
| GPT-4o | 0.662 | 0.590 | 0.756 | 0.709 | 0.840 | 0.824 | 0.723 |
| Gemini-3.1-Pro | 0.670 | 0.613 | 0.781 | 0.777 | 0.743 | **0.845** | 0.767 |
| GeoChat | 0.316 | 0.378 | 0.282 | 0.279 | 0.306 | 0.577 | 0.605 |
| LHRS-Bot | 0.532 | 0.221 | 0.295 | 0.316 | 0.242 | 0.189 | 0.325 |
| VILA1.5-8B (fine-tuned) | 0.811 | 0.687 | 0.792 | 0.959 | 0.882 | 0.814 | 0.772 |
| UrbanMLLM-3B | 0.803 | 0.677 | 0.778 | 0.946 | 0.859 | 0.816 | **0.782** |
| UrbanMLLM-8B | 0.822 | **0.693** | **0.795** | **0.963** | 0.879 | 0.827 | 0.776 |

attributed to the potential inclusion of similar indicator data within its training data, enabling it to perform strong inference in this special problem.

### 4.3. Ablation Analysis

To evaluate the effectiveness of UrbanMLLM's key architectural designs, we conducted an ablation study on different model variants, as presented in Table 5 and Table 6. Specifically, we assessed the performance of UrbanMLLM without the cross-view perceiver and without inter-

*Table 4.* Performance comparisons on cross-view imagery-based urban understanding tasks.

| Cross-View Task | IP | | | SRR |
|---|---|---|---|---|
| Subtasks | Med. income | Pov. ratio | Population | |
| QwenVL-2.5-7B | 0.521 | 0.603 | 0.510 | 0. 245 |
| VILA1.5-8B | 0.607 | 0.497 | 0.525 | 0.220 |
| Qwen3-VL-32B | 0.598 | 0.566 | 0.493 | 0.250 |
| InternVL-2-40B | 0.597 | 0.572 | 0.462 | 0.280 |
| Qwen-VL-Plus | 0.648 | 0.618 | 0.489 | 0.299 |
| GPT-4o | 0.684 | **0.848** | 0.499 | 0.322 |
| Gemini-3.1-Pro | **0.772** | 0.626 | 0.485 | 0.332 |
| VILA1.5-8B (fine-tuned) | 0.740 | 0.680 | **0.561** | 0.341 |
| UrbanMLLM-3B | 0.657 | 0.598 | 0.548 | 0.294 |
| UrbanMLLM-8B | 0.761 | 0.668 | 0.543 | **0.345** |

*Table 5.* Ablation study of UrbanMLLM variants on satellite imagery understanding tasks.

| Variants | SC | | OR | SRR | GL | IP | |
|---|---|---|---|---|---|---|---|
| | Single | Multi | | | | Pop | Nightlight |
| UrbanMLLM-8B | **0.787** | **0.776** | **0.800** | 0.664 | **0.871** | **0.872** | **0. 798** |
| w/o cross-view perceiver | 0.760 | 0.720 | 0.778 | 0.649 | 0.844 | 0.862 | 0.787 |
| w/o pre-training | 0.779 | 0.759 | 0.797 | **0.686** | 0.846 | 0.855 | 0.787 |

*Table 6.* Ablation study of UrbanMLLM variants on street view imagery understanding tasks.

| Variants | SC | OR | LM | SRR | GL | IP | |
|---|---|---|---|---|---|---|---|
| | | | | | | BF | WE |
| UrbanMLLM-8B | **0.822** | **0.693** | **0.795** | **0.963** | 0.879 | 0.827 | 0.776 |
| w/o cross-view perceiver | 0.822 | 0.679 | 0.753 | 0.948 | 0.852 | **0.834** | **0.778** |
| w/o pre-training | 0.811 | 0.687 | 0.792 | 0.959 | **0.882** | 0.814 | 0.772 |

leaved pre-training. Our results demonstrate that both the cross-view perceiver and interleaved pre-training contribute positively to overall model performance, confirming the effectiveness of each design. The removal of the cross-view perceiver leads to a larger performance degradation. Quantitatively, disabling the cross-view perceiver results in an average performance drop of 3.04% on satellite tasks (Table 5) and 1.54% on street-view tasks (Table 6). While we also observed some exceptions, for example, removing the cross-view perceiver slightly improved performance on street-view-based indicator prediction tasks. This suggests that for highly specific street-level attributes like the beautiful and wealthy level, the integration of macroscopic satellite knowledge might introduce noise, as these tasks primarily rely on intrinsic street-view knowledge.

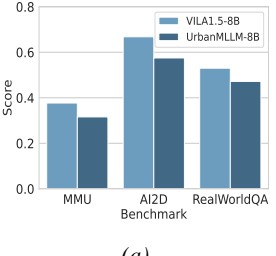

*(a)*

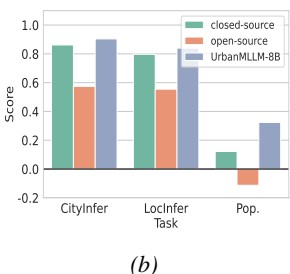

*(b)*

*Figure 4.* (a) Performance comparison of VILA1.5-8B, and UrbanMLLM on general benchmark. (b) Performance comparison of SOTA model and our model on Citybench tasks.

### 4.4. General Capabilities

We present evaluation results on general MLLM benchmarks in Figure 4a. As shown in Figure 4a, while UrbanMLLM performs well across various urban tasks, it also retains strong stability in general scenarios, including

MMMU (Yue et al., 2024), AI2D (Kembhavi et al., 2016), and RealWorldQA (XAI Organization, 2024). These results demonstrate that UrbanMLLM is competitive not only in general visual tasks but also in real-world spatial understanding and cross-view integrated reasoning, all of which are essential for comprehensive urban understanding.

We further validate UrbanMLLM's performance on City-bench (Feng et al., 2025b), an external urban understanding benchmark without additional training. We selected the City Inference (CityInfer), Location Inference (LocInfer), and Population Prediction (Population) tasks for evaluation, using Accuracy, Accuracy@25km, and $R^2$ as metrics. As shown in Figure 4b, UrbanMLLM consistently outperforms the reported state-of-the-art model. These results underscore the effectiveness and generalization ability of our proposed method across diverse urban understanding tasks.

## 5. Conclusion

In this paper, we introduce UrbanMLLM, a novel MLLM specifically developed for comprehensive urban understanding by jointly learning from satellite and street-view imagery. Leveraging a large-scale cross-view dataset for pre-training and an innovative cross-view perceiver architecture, UrbanMLLM effectively integrates complementary information from diverse views of urban appearance. Through extensive evaluation, our model consistently outperforms existing methods, demonstrating significant advancements across a wide range of urban understanding tasks. This work advances the use of web data and MLLMs for urban understanding, offering a scalable and versatile solution for understanding complex urban environments aimed at fostering equitable and resilient cities.

## Acknowledgements

This work was supported in part by the National Key Research and Development Program of China under Grant 2024YFC3307603. We thank the anonymous reviewers for their valuable and constructive feedback.

## Impact Statement

The satellite and street-level imagery utilized in UrbanM-LLM are sourced from public platforms, including Google Street View and Baidu Maps. These platforms employ automated masking to obscure sensitive visual details, such as faces and license plates. All visual data is used exclusively for academic research; furthermore, images are maintained at a coarse, region-level resolution to ensure that no individual-level visual content is exposed. The dataset is entirely de-identified, ensuring that no personally identifiable information (PII) is collected, stored, or annotated during the construction process.

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

# A. Appendix

## A.1. Implementations for Reproducibility

We perform experiments using Python 3.10 and PyTorch 2.3.0+cu121 with 8× NVIDIA A100 GPUs. Here we provide detailed values of the hyperparameters used in the experiments for reproducibility in Table 7 and Table 8 for the training and testing, respectively. We initialize our model's weights using the pretrained VILA-1.5, and adapt the AdamW optimizer with a cosine learning rate scheduler during training. The training process consists of two stages: in the first stage, we train on the entire interleaved pretraining dataset with a batch size of 8 for one epoch, corresponding to 7200 steps with 8 hours. For the second stage, we fine-tune the model on the SFT dataset at a batch size of 16 for one epoch with 8 hours.

## A.2. Baselines

We evaluated several advanced MLLMs and CLIP-based model on the our urban understanding benchmark. We add RemoteCLIP (Liu et al., 2024a) and OpenAICLIP (Radford et al., 2021) for comparison. But some of the MLLMs are not pretrained on multi-image data or do not support multi-image inference, such as LLaVA-Next (Liu et al., 2024b), so only single-image tasks are evaluated. For VILA-1.5 (Lin et al., 2024), InternVL2 (Chen et al., 2023), and QwenVL-2.5 (Wang et al., 2024a), the whole benchmark evaluation is done. In addition to the open-source models, state-of-the-art closed-source models Qwen-VL-Plus and GPT-4o are also fully evaluated on the benchmark. Specifically, we assess some satellite domain-specific models, GeoChat (Kuckreja et al., 2024) and LHRS-Bot (Muhtar et al., 2024), to further prove our capability. To ensure fairness, domain-specific models that are not yet open-source, such as UrbanCLIP (Yan et al., 2024), UrbanVLP (**?**), SkysenseGPT (Luo et al., 2024) and H$^2$RSVLM (Pang et al., 2024), are excluded from this evaluation. And we also evaluate the results comparison of fine-tuned VILA1.5-8B.

*Table 7.* Hyperparameter settings for training.

|                       | Stage1   | Stage2  |
| --------------------- | -------- | ------- |
| Optimizer             | AdamW    | AdamW   |
| Learning Rate         | 5e-5     | 1e-4    |
| Batch Size            | 8        | 16      |
| Accumulation Step(s)  | 1        | 2       |
| Weight Decay          | 0.0      |         |
| Epoch(s)/Step(s)      | 1 Epoch  | 1 Epoch |
| Save Steps            | 1200     | 750     |
| Scheduler             | Cosine   |         |
| Warmup Ratio          | 500      | 100     |
| Model Max Length      | 2048     |         |

## A.3. Dataset Details

### A.3.1. DATA COLLECTION

The images are primarily collected from two sources: Google Maps API (Google, 2024) for street view images, and ESRI for satellite imagery. For street view imagery collection, we randomly generate 2,000 random points in each census tract polygon and use their coordinates to query Google Maps API, returning street view image patches and real coordinates. We scrape over 2 million street view images and all satellite imagery of zoom level 15 across the United States. We further gather a variety of socioeconomic data of census tract and grid level from WorldPop, NIH and US government. We also collect a series of open datasets, such as Google Landmarks Recognition (Weyand et al., 2020), Visual Genome (Krishna et al., 2016) and Place Pulse (Dubey et al., 2016), etc. By applying some domain-specific adaptation to the original ones, we build a more well-rounded dataset. We use census tract boundary data in 2019 (Bureau, 2019), and gather street view and satellite images in 71,433 out of all 73,868 census tracts in the United States, which is about 96.7%. The Google street view images can be acquired using coordinate queries, however, we do not know the exact coordinate of where the street view exists. We randomly generate 2,000 points in each census tract and use these points to query street view images. This is a random process, thus we are not able to sample all the images in a census tract considering the time cost. In fact, in some less populated areas, it is quite hard to get street view images because the randomly generated query points in these areas are

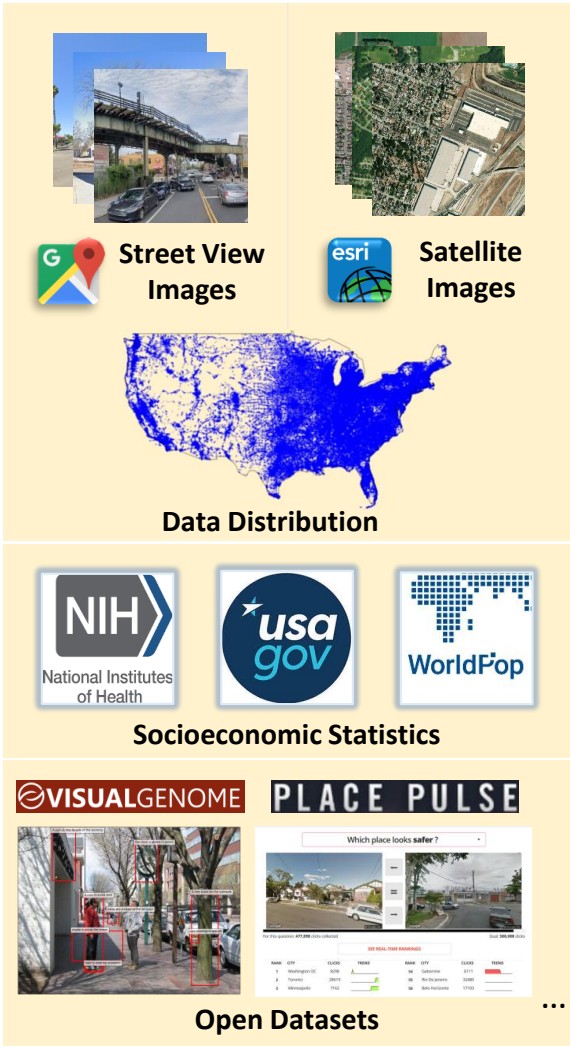

*Figure 5.* Data sources include imagery, socioeconomic statistics, and open datasets, with nationwide coverage.

*Table 8.* Hyperparameter settings for testing.

| Hyper-parameter | Value |
|---|---|
| Temperature | 0.2 |
| Top_p | None |
| Num Beams | 1 |
| Conv Mode | v1/llama_3 |
| Max New Tokens | 128 |

always off-road, which is also the main reason for the missing 3.3% coverage. In the end, we randomly sample about 200 images in each census tract, which have been proved to be effective in indicator prediction tasks.

### A.3.2. URBAN INTERLEAVED PRETRAINING DATASET

We first build a large-scale cross-view interleaved pretraining dataset. For each census tract, we match the coordinates between satellite and street view imagery. In order to control the size of inputs, at most 5 street view images are matched with a single satellite image. For the next step, we use a powerful open-source MLLM, InternVL2-40B, to generate detailed descriptive captions for them. Using a similar data structure in MMC4 (Zhu et al., 2024), the county name and coordinates

*Table 9.* Dataset and benchmark data size for different sources and tasks

| Source | Task | Dataset Size | Benchmark Size |
|---|---|---|---|
| Street View | Scene Classification (SC) | 30,000 | 1,000 |
| | Object Reasoning (OR) | 90,000 | 1,000 |
| | Landmark Recognition (LR) | 30,000 | 1,000 |
| | Spatial Relationship Reasoning (SRR) | 30,000 | 1,000 |
| | Geo-Localization (GL) | 30,000 | 1,000 |
| | Indicator Prediction (IP) | 90,000 | 3,000 |
| Satellite | Scene Classification (SC) | 51,759 | 8,668 |
| | Object Reasoning (OR) | 115,115 | 5,556 |
| | Spatial Relationship Reasoning (SRR) | 90,000 | 8,250 |
| | Geo-Localization (GL) | 29,629 | 1,000 |
| | Indicator Prediction (IP) | 60,000 | 2,000 |
| Cross-View | Spatial Relationship Reasoning (SRR) | 30,000 | 1,000 |
| | Indicator Prediction (IP) | 120,000 | 4,000 |

of the satellite image are also embedded to the interleaved pre-training data, together with imagery and caption embeddings.

We use a Human-AI mixture method for pre-training caption quality validation. We first use two powerful open-source MLLMs, VILA-1.5-40B and LLaVA-Next-34B to judge if the caption matches with the image. If either of them thinks it is not a match, we will proceed to send this case to GPT-4o, which has state-of-the-art comprehension ability, but not quite affordable for large-scale deployment. If GPT-4o also thinks there is a problem with the case, a graduate-level human being will manually check this case to give the final judgment. In order to quantify the caption quality improvement, we further use GPT-4o to regenerate captions for excluded images with human assistance to test the quality improvement. We use CLIP-Score as the evaluation metric and calculate our original caption score and cleaned caption score of 10,000 samples, resulting in 29.99535 and 29.99571 respectively. As a matter of fact, the original caption quality is good enough and only about 1.3 out of a thousand images are marked as unmatched by two-stage MLLM verification, and the regeneration process enhances the caption quality only by 0.0012%.

Then we construct the instruction tuning dataset, which is categorized into 3 major types: satellite-only, street view-only, and cross-view data. Street view images offer ground-level data of various environments, including urban and rural areas. These images provide detailed features of the environment. In contrast, satellite imagery complements this by providing top-down views, capturing an overall perception of the entire landscape. In the dataset, not only satellite and street view images linked respectively with diverse tasks such as scene classification, object reasoning, spatial relationship reasoning, cross-view combinational tasks of socioeconomic prediction and image retrieval are delicately designed to further enhance MLLM's comprehensive understanding of urban environment.

In the construction of dataset, a lot of street view tasks are lacking groundtruth labels. Therefore, we use the light-weight object detection specialized model to generate groundtruth bounding boxes for object reasoning tasks, and use powerful open-source MLLMs to identify the scene class and spatial relationship of the street view image. For geo-localization tasks, we simply use latitude and longitude of the images to match the boundaries of census tracts in the United States.

A.3.3. Urban Instruction Tuning Dataset

As shown in the Figure 6, we construct an instruction tuning dataset for a variety of urban tasks, ranging from perception, reasoning to numerical prediction. The data size for each task of the dataset and benchmark is listed in Table 9. To clarify the provenance of labels used in training and evaluation, we add a task-level provenance summary in Table 10. The table explicitly separates SFT supervision from evaluation targets and distinguishes human annotation, detector-derived labels, MLLM-generated annotations, rule-based labels, POI/GPS matching, open-source annotations, and human verification or post-editing. Importantly, detector and MLLM outputs are mainly used as bootstrapping signals for SFT, whereas evaluation targets are primarily based on rules, metadata/GPS-POI matching, open-source annotations, or human-checked labels. Therefore, the reported gains should not be interpreted as arising from self-generated test labels.

*Table 10.* Task-level provenance of SFT supervision and evaluation targets. The table separates supervision used for instruction tuning from labels used for evaluation, and distinguishes human-checked labels, detector-derived labels, MLLM-generated annotations, rule-based labels, POI/GPS matching, and open-source annotations.

| View | Task | SFT Supervision | Evaluation Target |
|------|------|-----------------|-------------------|
| Street View | SC | MLLM annotation + human check | Human-checked labels |
| Street View | OR | GroundingDINO | Detector-derived labels + verification |
| Street View | LR | POI/GPS matching | POI/GPS matching |
| Street View | SRR | Rule-based labels + LLM rewrite | Rule-based labels |
| Street View | GL | Rule-based labels | Rule-based labels |
| Street View | IP | Rule-based labels | Rule-based labels |
| Satellite | SC | Open-source annotations | Open-source annotations |
| Satellite | OR | Open-source annotations | Open-source annotations |
| Satellite | SRR | Open-source annotations | Open-source annotations |
| Satellite | GL | Rule-based labels | Rule-based labels |
| Satellite | IP | Rule-based labels | Rule-based labels |
| Cross-View | SRR | Rule-based labels | Rule-based labels |
| Cross-View | IP | Rule-based labels | Rule-based labels |

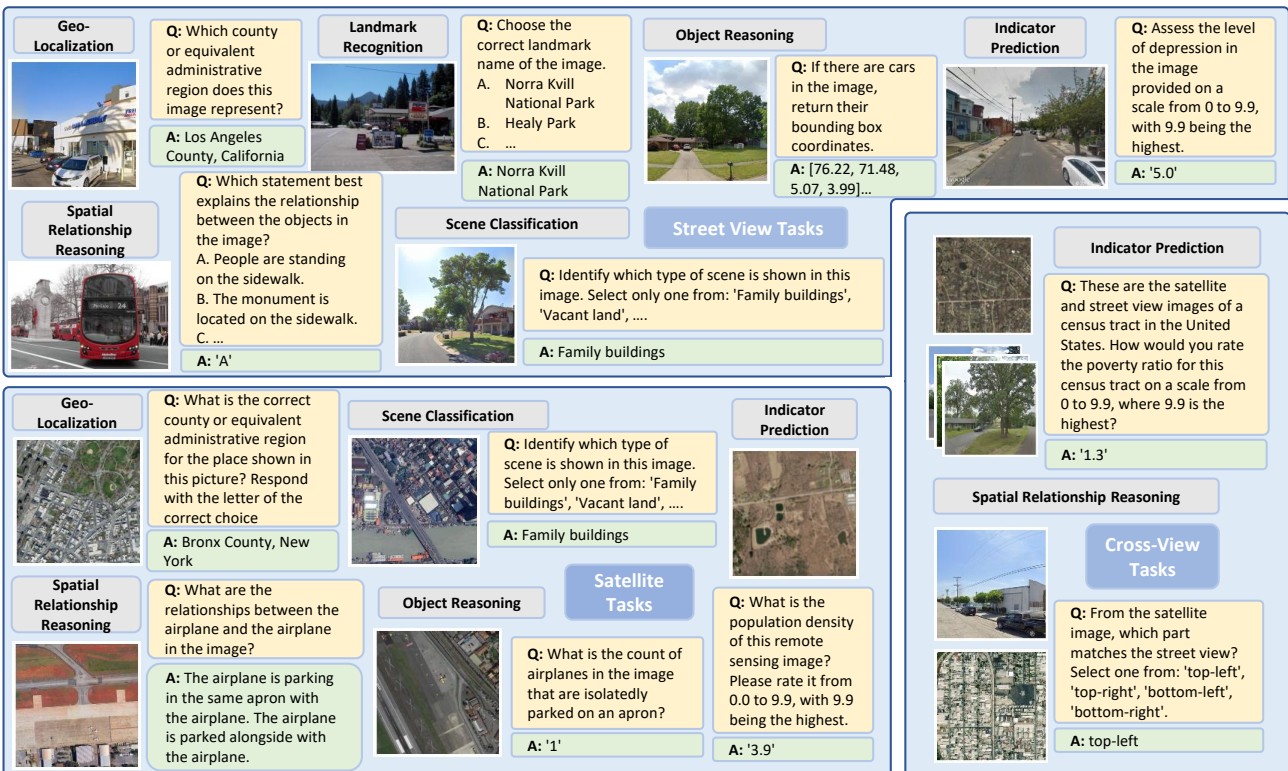

*Figure 6.* Examples of satellite, street view, and cross-view tasks in instruct tuning dataset. Diverse task categories include Scene Classification (SC), Object Reasoning (OR), Landmark Recognition (LR), Spatial Relationship Reasoning (SRR), Geo-Localization (GL) and Indicator Prediction (IP).

## A.4. Experimental Results

### A.4.1. DATA SCALE STUDY

We provide the results of UrbanMLLM-8B trained on different dataset scales, including 0.01 million, 0.10 million, 1.86 million, and 4.50 million images (Table 11). The cross-view SPR performance increases from 0.319 to 0.345 with the increasing data scale. These findings indicate that while increasing data volume yields measurable performance gains, the rate of improvement diminishes at larger scales, suggesting a potential saturation effect in data scaling efficiency.

*Table 11.* Cross view imagery-based urban understanding results with different data scale on two tasks.

| CV | IP | | | SRR |
|---|---|---|---|---|
| Sub-task | Med. income | Pov. ratio | Pop. | |
| 0.01M | 0.740 | 0.649 | 0.543 | 0.319 |
| 0.10M | 0.747 | 0.671 | 0.559 | 0.291 |
| 1.86M | 0.757 | **0.678** | **0.568** | **0.348** |
| 4.50M | **0.761** | 0.668 | 0.543 | 0.345 |

*Table 12.* Image number ablation study of UrbanMLLM on cross-view imagery tasks.

| Number | IP | | |
|---|---|---|---|
| | Med. income | Pov. ratio | Population |
| 2 | 0.766 | 0.696 | 0.596 |
| 4 | 0.789 | 0.689 | 0.596 |
| 6 | **0.809** | **0.720** | **0.614** |

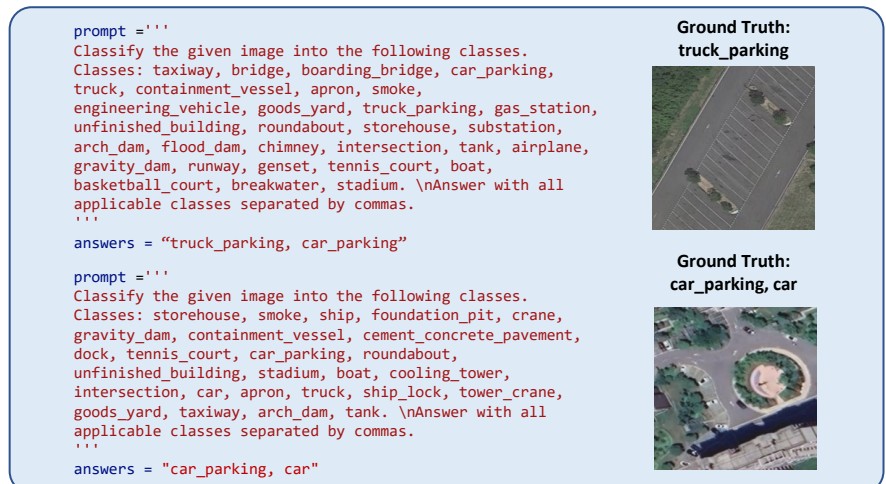

*Figure 7.* Bad case on scene classification.

### A.4.2. ABLATION STUDY

As our model supports multi-image as input, we conduct an ablation study on the number of images (N = 2, 4, 6) for indicator prediction tasks. For example, 6 images means one satellite image and five street-view images as input. As shown in the Table 12, it can be seen that more images bring a certain performance gain for the indicator prediction task.

### A.4.3. CASE STUDY

We show some examples of bad cases in scene classification and indicator prediction tasks. The results are shown in Figure 7, 8, 9. Firstly, in the scene classification task, the model misclassifies the image with a truck parking as a car parking. Although there are a few differences between the two classes, the more granular understanding of the urban environment is required to distinguish them. Secondly, in the indicator prediction task, as shown in Figure 8, 9, the model predicts the population density of an urban area as 6.8 and the actual value is 9.9 using a satellite image. The model fails to capture detailed information with a single-view image, which makes it challenging for the model to learn from the limited dataset. For poverty rate prediction, the model gets a high score of 5.4, but the poverty rate is 2.6. It's may be due to the number of street view images in the dataset is not enough to learn the detailed information of the urban environment. We also present

```
prompt ='''
Please rate the population density of this image from 0.0
to 9.9, with 9.9 being the highest. Only output the score
'X.X'.
'''
answers = "6.8"
```

**Ground Truth: 9.9**

```
prompt ='''
Assess the level of wealth in the image provided on a scale
from 0 to 9.9, with 9.9 being the highest. Only output the
score. Example format: 'X.X'.
'''
answers = "3.3"
```

**Ground Truth: 8.0**

*Figure 8.* Bad case on indicator prediction task in single view imagery understanding.

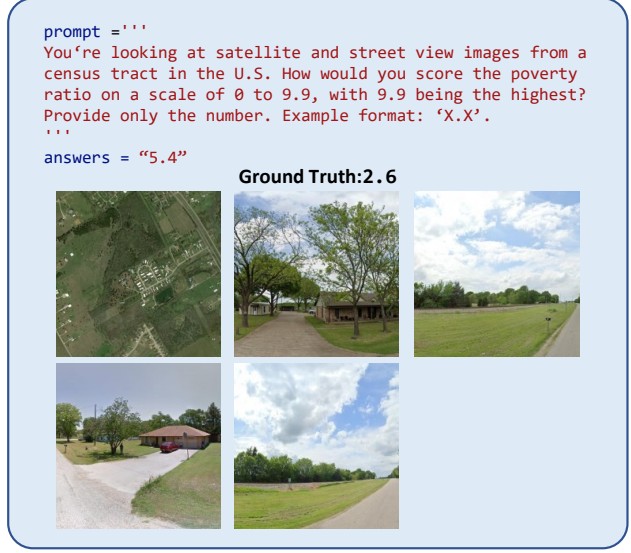

```
prompt ='''
You're looking at satellite and street view images from a
census tract in the U.S. How would you score the poverty
ratio on a scale of 0 to 9.9, with 9.9 being the highest?
Provide only the number. Example format: 'X.X'.
'''
answers = "5.4"
```

**Ground Truth:2.6**

*Figure 9.* Bad case on indicator prediction task in cross view imagery understanding.

the performance of various tasks across different models, as illustrated in Figure 10 to  13. The results demonstrate that our model exhibits a clear advantage in urban understanding tasks.

## A.5. Limitations and Future Work

Although UrbanMLLM covers many urban understanding tasks and achieves strong performance, it still has several limitations. First, our dataset and main evaluations are mainly based on U.S. cities. Although the results on CityBench show that the model can transfer to some non-U.S. scenarios, more evaluations across different countries, city layouts, and geographic regions are needed to better understand its generalization ability. In future work, we plan to expand the dataset to more regions and further pre-train the model to improve its global applicability.

Second, part of the supervision used in the SFT pipeline is generated by detectors or existing MLLMs. This makes data construction more scalable, but these labels are not as reliable as fully human-verified annotations and may contain noise or bias. Third, in single-view settings, using duplicate images as a fallback is only a practical implementation choice. It does

not directly evaluate the model's ability to fuse information from different views. Future work will focus on improving data quality, designing more rigorous multi-view evaluation protocols, and incorporating additional modalities such as temporal street-view sequences, map layers, and geographic metadata.

### A.6. Ethical Statement

Our model leverages a large volume of satellite and street-view imagery, which raises potential concerns regarding individual privacy. While the resolution of the satellite images is insufficient to identify individuals, it can still capture environmental changes associated with human activity. The street-view images were crawled and downloaded from the Google platform, where sensitive information is already blurred to ensure that no private data is compromised. To respect the intellectual property rights of major commercial platforms, the dataset we plan to release will include only the associated metadata. Publicly available tools can use this metadata to retrieve the original images from their official sources, thereby avoiding the direct redistribution of proprietary content.

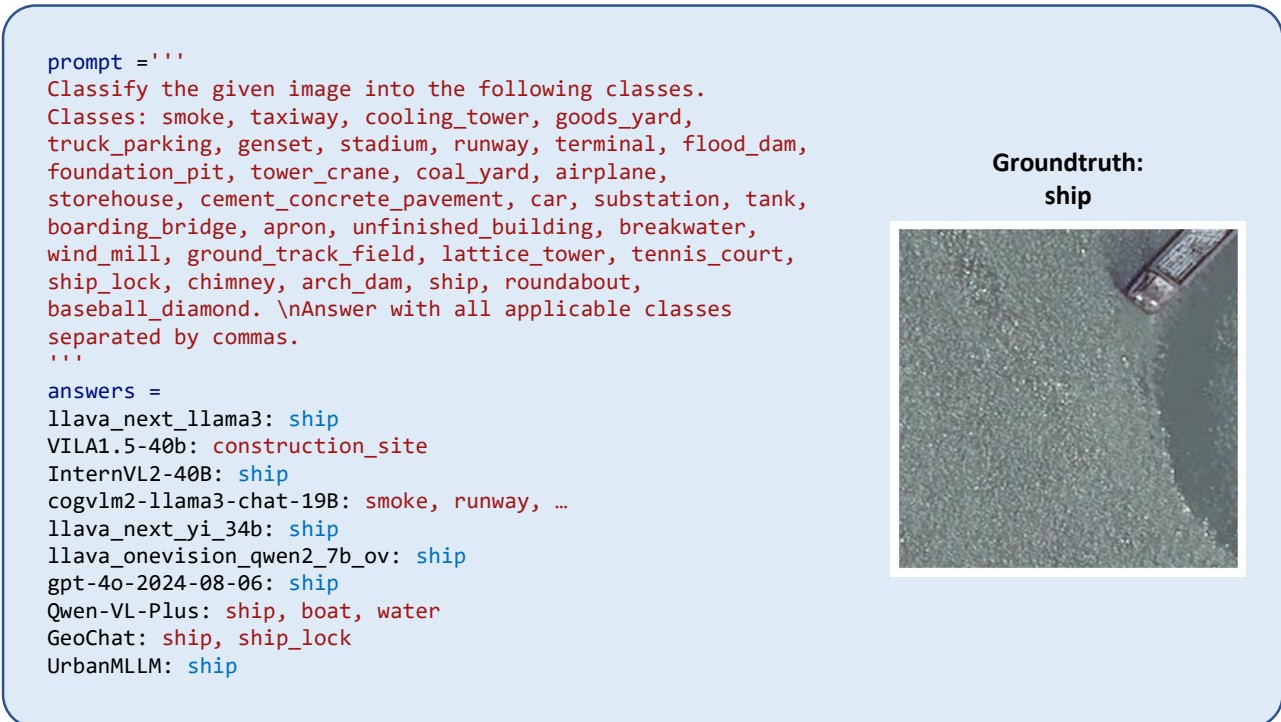

*Figure 10.* Illustration of the satellite image scene classification task.

```
prompt ='''
Which scene category does this image fit into? Choose just
one from: 'Family buildings', 'Mixed residential and
commercial buildings', 'Commercial and office buildings',
'Industrial and manufacturing', 'Transportation and
utility', 'Public facilities and institutions', 'Open space
and outdoor recreation', 'Vacant land', 'Unknown'. Reply
with only one of the quoted options.
'''

answers =
llava_next_llama3: Transportation and utility
VILA1.5-40b: Mixed residential and commercial buildings
InternVL2-40B: 'Transportation and utility'
cogvlm2-llama3-chat-19B: Mixed residential and commercial
buildings
llava_next_yi_34b: 'Transportation and utility'
llava_onevision_qwen2_7b_ov: Transportation and utility
gpt-4o-2024-08-06: "Family buildings"
Qwen-VL-Plus: 'Mixed residential and commercial buildings'
GeoChat: 'Family buildings'
UrbanMLLM: Transportation and utility
```

**Groundtruth:**
**Transportation and utility**

*Figure 11.* Illustration of the street-view image scene classification task.

```
prompt ='''
From the options below, which county or administrative
region is depicted in this image? Submit only the letter of
the correct choice.
A. Ventura County, California
B. Salt Lake County, Utah
C. Baltimore County, Maryland
D. Maricopa County, Arizona
Answer only with A, B, C, or D, without any additional text.
Example output: 'A'
'''

answers =
llava_next_llama3: D
VILA1.5-40b: A
InternVL2-40B: B
cogvlm2-llama3-chat-19B: A
llava_next_yi_34b: D
llava_onevision_qwen2_7b_ov: B
gpt-4o-2024-08-06: B
Qwen-VL-Plus: A
GeoChat: D
UrbanMLLM: B
```

**Groundtruth:**
**B**

*Figure 12.* Illustration of the satellite image geo-localization task.

```
prompt ='''
What is the correct county or equivalent administrative
region for the place shown in this picture? Respond with
the letter of the correct choice.
A.  Cook County, Illinois
B.  Allegheny County, Pennsylvania
C.  Baltimore City, Maryland
D.  Jefferson County, Alabama
Answer only with A, B, C, or D, without any additional text.
Example output: 'A'
'''
answers =
llava_next_llama3: B
VILA1.5-40b: A
InternVL2-40B: B
cogvlm2-llama3-chat-19B: B
llava_next_yi_34b: A
llava_onevision_qwen2_7b_ov: B
gpt-4o-2024-08-06: B
Qwen-VL-Plus: B
GeoChat: A
UrbanMLLM: B
```

**Groundtruth: B**

*Figure 13.* Illustration of the street-view image geo-localization task.

```
prompt ='''
You are a powerful street-view image captioner.
Please create captions describing the contents of the given
image.
The caption annotation procedure follows the principles of:
(1): Describing object attributes, including object
quantity, color, material, shape, size, and spatial
position (including absolute position in the image and
relative position between objects);
(2): The annotation process involves just describing
specific object;
(3): Instead of describing the imaginary content, only
describing the content one can determine confidently from
the image.
Do not describe the contents by itemizing them in list form.
Minimize aesthetic descriptions as much as possible;
(4): Please output less 35 words.
'''

Answer ='''
A three-story residential building with light green and
beige exterior, white trim, and multiple windows. A red SUV
and a black car parked on the street. A small white garage
and a tree in the background.
'''
```

**Captioner: InternVL2-40B**

*Figure 14.* Illustration of the satellite image captioning task.

```
prompt ='''
You are a powerful remote sensing and aerial image
captioner.
Please create SHORT captions describing the contents of the
given image.
The caption annotation procedure follows the principles of:
(1): Describing object attributes, including object
quantity, color, material, shape, size, and spatial
position (including absolute position in the image and
relative position between objects);
(2): The annotation process involves just describing the
overall scene of the image and some specific object;
(3): Instead of describing the imaginary content, only
describing the content one can determine confidently from
the image.
Do not describe the contents by itemizing them in list form.
Minimize aesthetic descriptions as much as possible;
(4): Please output within 25 words.
'''

Answer ='''
Aerial view of a suburban area with a mix of commercial
buildings, parking lots, and green spaces. A major road
curves through the scene, with a large white building near
the center.
'''
```

**Captioner: InternVL2-40B**

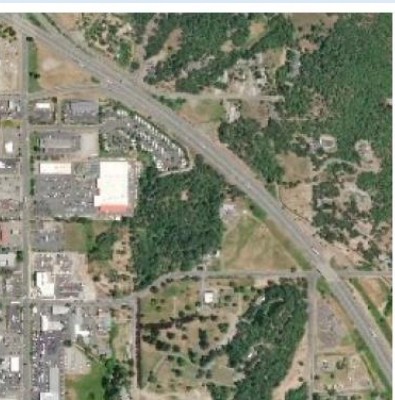

*Figure 15.* Illustration of the street-view image captioning task.

