# OpenReview forum: "UrbanMLLM: Joint Learning of Cross-view Imagery for Urban Understanding"
_ICML.cc/2026/Conference — ICML 2026 regular_

### Official Review · Reviewer_f8xp · 2026-03-11

**Soundness:** 3
**Presentation:** 3
**Significance:** 3
**Originality:** 2
**Overall Recommendation:** 4
**Confidence:** 4

**Summary:**

This paper proposes UrbanMLLM, a unified MLLM for urban understanding, which aims to jointly model the complementary relationship between satellite remote sensing imagery  and street view imagery. The authors constructed a large-scale data resource covering the United States, as well as instructional fine-tuning data covering a variety of urban tasks.

**Compliance With Llm Reviewing Policy:**

Affirmed.

**Final Justification:**

The paper is well-motivated and presents a reasonable cross-view MLLM framework with strong empirical results and a valuable large-scale dataset. My initial concerns were about generalization, limited methodological novelty, and data construction rigor. The rebuttal addresses these to a reasonable extent, particularly by clarifying evaluation on CityBench and the multi-stage data validation pipeline. I maintain my recommendation as Weak Accept.

**Key Questions For Authors:**

Will prompts used for caption/QA generation and cleaning be released simultaneously? This would significantly enhance the paper's community impact and value.

**Limitations:**

yes

**Strengths And Weaknesses:**

Strengths
- The method design aligns with the motivation: the cross-view perceiver addresses the problem of information isolation caused by independent encoding of features from different viewpoints, and staggered pre-training further strengthens cross-image contextual relationships, resulting in a reasonable overall framework.
- The empirical support is relatively complete: quantitative results are provided for satellite, street view, and cross-view tasks, directly supporting the main claims.
- The dataset constructed in the paper is large: 4.5M cross-view image pairs; 700K instruction data. Such large-scale data resources are relatively rare in urban multimodal research, thus the dataset itself has considerable value.
- The training setup and implementation details are provided in the appendix with specific hyperparameters, hardware, and training time information, facilitating understanding of the training cost and reproducibility threshold.

Weaknesses
- The paper does not provide sufficient evidence that the model generalizes beyond US cities. Additional evaluation on non-US regions would strengthen the claims.
- The paper's innovation mainly lies in: dataset construction; and application scenario expansion. The model method itself is primarily based on existing technologies, therefore the algorithmic innovation is relatively limited.
- Information regarding data construction is scattered throughout the main text and appendix, and there's a slight inconsistency between "matching 3–6 street views per satellite" and the appendix's "maximum 5," which needs clarification or explanation (training/inference cap vs. original number of matches).
- The proposed data construction method is excellent. For a two-stage MLLM data cleaning method, a 0.13% error rate is tolerable. Could you provide specific examples of errors? Furthermore, the data cleaning method used in the paper can be understood as using the model to generate data and then using the model to validate the data again, which is not considered rigorous validation in data quality assessment. The reported improvement in CLIP score (29.99535 → 29.99571) is extremely small, making it difficult to assess the practical impact of the proposed cleaning procedure.
- The scale of manual validation is also too small, lacking information on: sample size; inter-annotator agreement; annotation standards.

---

> ### Author Rebuttal · Authors · 2026-03-31
>
> Thank you for the positive and careful review.
>
> ### W1. Generalization beyond US cities
>
> Our evidence is not limited to in-domain testing: we already report results on **CityBench**, which covers **13 cities worldwide** (Fig. 4(b)). On this benchmark, UrbanMLLM outperforms both the strongest **closed-source** and **open-source** baselines. Given the substantial variation in urban form and geographic context across these cities, we view this as meaningful evidence of **cross-city transfer beyond the training distribution**.
>
>
> ### W2. Novelty of the method
>
> We respectfully believe the contribution is not limited to dataset construction. The methodological novelty lies in designing an architecture specifically for **cross-view urban understanding**.
>
> Our **cross-view perceiver** enables **mutual interaction between satellite and street-view features** through **cross-attention fusion with adaptive gating**, addressing the common problem that features from different views remain isolated. This differs from general-purpose MLLMs such as **Qwen-VL, CogVLM,** and **Gemma**, which mainly focus on **image-text fusion**, and from prior urban models such as **UrbanVLP**, which mainly use **feature concatenation** or shallow fusion. Our design instead targets **cross-view alignment and relational grounding** at the feature level. We will clarify this distinction more explicitly in the revision.
>
> ### W3. Clarification on street-view pairing consistency
>
> The dataset itself is consistent; the inconsistency is only textual. Each sample contains **1 satellite image** matched with **up to 5 street-view images**, i.e., **6 images in total**. The phrase **“3-6 street views”** is a typographical error rather than a difference between training and inference settings. We will correct this in the revision.
>
> ### W4. Data construction and cleaning rigor
>
> Our cleaning pipeline should not be understood as a model generating data and then validating itself. Instead, it is a **multi-model, disagreement-driven validation framework**. Captions are first generated, then checked independently by **VILA-1.5-40B** and **LLaVA-Next-34B**; flagged cases are re-generated/reviewed by **GPT-4o**; persistent disagreements are finally resolved by a **graduate-level human annotator**. Thus, MLLMs provide **scalable screening and triage**, while **human review remains the final safeguard**.
>
> This distinction matters because many errors are **semantically important but weakly reflected in CLIP**. For example, a street-view image of a **quiet suburban street** may be incorrectly captioned as **“a densely populated urban area with numerous houses.”** Such a caption may still overlap with the image at a coarse lexical level, so CLIP changes little, but it distorts **density, scene type, and spatial context**, which are critical for **urban scene classification, cross-view alignment, and spatial reasoning**. This is representative of the typical errors we observed. For this reason, **CLIP is only an auxiliary metric**, not the main evidence for cleaning effectiveness.
>
> ### W5. Manual validation scale and annotation protocol
>
> The reported **100-sample manual check** is a **supplementary sanity check**, not the primary basis of our quality claim. Its purpose is to assess **caption descriptiveness**, using a concrete criterion: whether the caption covers the **top-3 salient objects** and avoids obvious hallucination or mismatch. All 100 sampled cases satisfy this criterion.
>
> Our main evidence for reliability comes from the broader **multi-stage validation pipeline**, under which only about **0.13%** of image-caption pairs are ultimately identified as mismatched. We believe this residual error rate is the more relevant signal for dataset quality at scale.
>
> Regarding **sample size, annotation standards, and inter-annotator agreement**, we agree these should be described more explicitly. The human stage is used primarily for **adjudication of disputed cases**, rather than as a standalone large-scale multi-annotator benchmark. In the revision, we will make the **manual sample size, annotation criteria, and annotator protocol** explicit so that the role of human validation is clear and not overstated.
>
> ### Q1. Release of prompts
>
> Yes. We have already provided a **representative prompt example** in the anonymous repository, and we plan to release the **full prompts for caption/QA generation and cleaning** in the final public release. We will also include additional prompt details in the appendix to improve transparency and reproducibility.

---

> > ### Author Rebuttal · Reviewer_f8xp · 2026-04-02
> >
> > Thank you for the rebuttal. The clarifications on CityBench, the cross-view novelty, and the data cleaning pipeline address most of my concerns. I still encourage the authors to describe the data construction/validation process more clearly in the paper and to acknowledge the limited non-U.S. evaluation more explicitly. Overall, I will maintain my score.

---

> > > ### Author Response · Authors · 2026-04-04
> > >
> > > Thank you for the thoughtful follow-up and for maintaining your score. We are glad that our clarifications on CityBench, the cross-view novelty, and the data cleaning pipeline addressed most of your concerns. In the revision, we will expand the description of the data construction and validation pipeline and more explicitly state the current limitation in non-U.S. evaluation. We appreciate the constructive feedback.

---

### Official Review · Reviewer_zMBQ · 2026-03-11

**Soundness:** 3
**Presentation:** 2
**Significance:** 3
**Originality:** 3
**Overall Recommendation:** 4
**Confidence:** 4

**Summary:**

The paper addresses urban understanding from paired satellite and street-view imagery. It presents UrbanMLLM, a domain-specific multimodal model intended to combine broad spatial context from overhead imagery with local visual evidence from street-level views. The method centers on a cross-view perceiver that lets the two visual streams interact during encoding, together with an interleaved pre-training strategy built on cross-view image-text "urban documents." The authors also assemble a large U.S.-focused dataset containing matched cross-view images, captions, geolocation metadata, and instruction-style supervision. Experiments cover 13 tasks across satellite-only, street-view-only, and cross-view settings, and the paper reports generally improved performance over a range of open-source and commercial baselines.

**Compliance With Llm Reviewing Policy:**

Affirmed.

**Final Justification:**

I thank the authors for their thorough engagement throughout the review process. After carefully reconsidering the rebuttal, I have decided to raise my score to 4 (Weak Accept).
The dataset and evaluation framework are genuine contributions to the community, and the cross-view perceiver is a well-motivated design. The authors' responses have addressed my main concerns to a reasonable extent.  Some minor concerns remain, including the unexplained non-monotonic behavior in Table 10 and the label quality of the OR evaluation, but I consider these addressable in a revision rather than fundamental obstacles.
Overall, the authors have demonstrated sufficient scientific rigor in their responses, and I believe the paper makes a meaningful contribution to urban understanding with multimodal models.

**Key Questions For Authors:**

Q1. Can the authors provide a precise provenance table for each benchmark task, separating human annotation, detector-derived labels, MLLM-generated labels, and post-verification — covering both SFT training labels and evaluation targets separately? Appendix A.2.3 confirms that object detection models and MLLMs generate ground truth for several street-view tasks. If a substantial share of evaluation targets is model-derived, the interpretation of the gains changes materially.

Q2. How do the authors address the potential training-test leakage in geo-localization tasks? Geographic coordinates are embedded in the pre-training corpus, while GL is also a key evaluation task. Can the authors demonstrate that GL performance reflects genuine visual understanding—for example, by evaluating on a held-out set of census tracts unseen during pre-training, or by ablating the coordinate embeddings?

Q3. Can the authors explain why "w/o pre-training" in Table 5 produces scores identical to VILA1.5-8B (fine-tuned), and clarify the marginal contribution of interleaved pre-training? If the two variants are numerically equivalent, the interleaved pre-training stage may not be contributing as claimed, or the comparison is not properly controlled.

Q4. Can the authors justify the duplicate-image fallback in the cross-view perceiver for single-view inputs, and clarify its effect on the ablation results in Tables 5 and 6? If the perceiver performs self-attention over an identical image pair in single-view mode, the "w/o cross-view perceiver" baseline and the full model are not architecturally equivalent for these tasks, making it hard to attribute performance differences specifically to cross-view fusion.

Q5. What evidence is there that the model generalizes beyond the U.S.-centric training distribution, and can the authors provide continuous metrics (e.g., MAE, Pearson correlation) for indicator prediction tasks? The current "accuracy within margin of 2" metric permits approximately ±20% error on a 0–9.9 scale, which is coarse for socioeconomic prediction. Even a limited geographic transfer experiment, together with finer-grained regression metrics, would substantially strengthen the paper's broader claims.

**Limitations:**

The paper includes a limitations section (Appendix A.4), though it acknowledges only the geographic restriction. The issues raised above—particularly data leakage, synthetic supervision, and the degenerate single-view architecture—are not discussed as limitations.

**Strengths And Weaknesses:**

Strengths

S1. Many urban understanding tasks benefit from combining overhead spatial context with street-level visual evidence, whereas much prior work emphasizes only one view. The problem motivation is well-articulated and the gap with respect to existing urban MLLMs is clearly identified.

S2. Building a large paired satellite/street-view corpus with instruction-style supervision across 71,433 U.S. census tracts is a significant effort and may be useful to the community beyond this particular model.

S3. The paper covers satellite-only, street-view-only, and cross-view tasks across 13 diverse settings, giving a more informative picture of the system than a narrowly scoped benchmark would.

S4. The proposed cross-view perceiver design is well-motivated for the problem setting, and the ablations provide at least some evidence that both the fusion module and the interleaved training setup contribute positively.

Weaknesses

W1. In Table 5, the "w/o pre-training" variant appears to match the scores reported for VILA1.5-8B (fine-tuned) across all subtasks, suggesting the marginal gain from interleaved pre-training is smaller than the paper's narrative implies. More broadly, the ablations do not control for data scale versus architecture simultaneously, so it remains unclear how much improvement is attributable specifically to the cross-view perceiver versus the larger training corpus. The non-monotonic behavior in Table 10—where scaling from 1.86M to 4.50M images decreases performance on two metrics—further complicates the data scaling argument.

W2. Section 3.3 states that in single-view settings the cross-view perceiver duplicates the same image as both inputs, introducing an artificial self-attention bias. This means the "w/o cross-view perceiver" and full-model variants are not architecturally equivalent in single-view tasks, confounding the ablation results in Tables 5 and 6. Additionally, the entire dataset is U.S.-centric, and no transfer analysis or out-of-distribution evaluation is provided, limiting the scope of the paper's broader claims.

W3. Appendix A.2.3 reveals that geographic coordinates (longitude, latitude, county name) are directly embedded into the interleaved pre-training data. Since Geo-Localization (GL) is simultaneously a core evaluation task, it is unclear whether the model has genuinely learned visual geo-localization or is partly memorizing coordinate-to-region mappings seen during pre-training. This is the most pressing validity concern in the paper.

W4. A central concern is that a substantial portion of SFT training labels are model-generated: Appendix A.2.3 explicitly states that object detection models and open-source MLLMs are used to generate ground-truth labels for object reasoning, scene classification, and spatial relationship tasks. This is distinct from, and in addition to, the question of benchmark provenance. Gains that appear as model improvements may partially reflect alignment with the same model family used to generate supervision.

---

> ### Author Rebuttal · Authors · 2026-03-31
>
> Thank you for the constructive feedback.
> ### Q1/W4: Provenance of Supervision and Evaluation Labels
> We add a task-level provenance table in the revision / GitHub that explicitly separates SFT supervision from evaluation targets, and distinguishes human annotation, detector-derived labels, MLLM-generated labels, and human verification/post-editing. Importantly, detector/MLLM outputs are used mainly as bootstrapping signals for SFT, whereas evaluation targets are primarily based on rules, metadata/GPS-POI matching, open-source annotations, or human-checked labels. Therefore, the reported gains should not be interpreted as arising from self-generated test labels.
>
> |View|Task|SFT supervision|Evaluation target|
> |---|---|---|---|
> |Street View|SC|MLLM annotation+human check|human-checked labels|
> |Street View|OR|GroundingDINO|detector-derived+verification|
> |Street View|LR|POI/GPS matching| POI/GPS matching|
> |Street View|SRR|rule-based+LLM rewrite|rule-based|
> |Street View|GL|rule-based|rule-based|
> |Street View|IP|rule-based|rule-based|
> |Satellite|SC|open-source| open-source|
> |Satellite|OR|open-source|open-source|
> |Satellite|SRR|open-source|open-source|
> |Satellite|GL|rule-based|rule-based|
> |Satellite|IP|rule-based|rule-based|
> |Cross-View|SRR| rule-based|rule-based|
> |Cross-View|IP|rule-based|rule-based|
> ### Q2/W3: Train-Test Leakage in Geo-Localization
> We clarify that our GL evaluation is designed to avoid train-test leakage: test images are strictly unseen during training, and their associated metadata are also kept disjoint from the training set. In addition, although geographic coordinates appear in parts of the pre-training corpus, they are not provided as direct inputs during GL evaluation. Our strong results on CityBench, spanning 13 cities worldwide, further provide complementary evidence that the model captures transferable visual-geographic cues rather than dataset-specific leakage.
> ### Q3/W1: Contribution of Interleaved Pre-training
> The two variants are identical by design: "w/o pre-training" uses the same VILA1.5-8B backbone and the same downstream fine-tuning pipeline, but removes our interleaved pre-training stage. Our point is not that the gain comes from corpus scale alone. Rather, the cross-view perceiver was specifically designed for interleaved satellite-street-view pre-training, so the two components should be understood as a joint cross-view learning design. The perceiver provides the mechanism for aligning complementary views, while the interleaved corpus provides the supervision signal that makes such alignment learnable. Their contribution is reflected by the gap from w/o pre-training to UrbanMLLM-8B, especially on SRR (0.321->0.345), which most directly measures cross-view reasoning.
> ### Q4/W2: Duplicate-Image Fallback
> The key point is that the duplicated-image setting is not used to prove cross-view fusion helps single-view tasks. It is only a compatibility fallback so one implementation can handle both single-view and paired-view inputs. We will make this explicit and revise the discussion of Tables 5 and 6: for single-view tasks, the comparison reflects the effect of this implementation choice, not pure cross-view fusion. The cross-view perceiver should be judged mainly on paired-view tasks, where two genuinely different views are available. Please also refer to our response to Q2 for Reviewer PWy9.
> ### Q5/W2: Generalization and Finer-Grained Regression Metrics
> We provide two pieces of evidence for generalization beyond the U.S.-centric training distribution. First, we have already included results on CityBench (Fig. 4(b)), which covers 13 cities worldwide. Our model outperforms both the strongest closed-source and open-source baselines on this benchmark, indicating that the learned representation transfers beyond the training distribution.
>
> Second, we add finer-grained regression metrics for indicator prediction. Due to space limitations, we report results against the strongest baseline and include representative continuous metrics. Specifically, we report R2 / RMSE as follows:
>
> |Model| Pop.|Nightlight|Med.income| Pov.ratio|
> |---|---|---|---|---|
> |VILA1.5-8B (fine-tuned)|0.582/1.837 |0.366/2.323 |0.268/2.420|0.008/2.774|
> |Ours|0.605/1.794|0.378/2.290|0.355/2.272|0.013/2.754|
>
> These results show that the advantage of our model is preserved under continuous evaluation, beyond the coarse accuracy within margin of 2 metric.
> ### Limitations
> First, some supervision in the SFT pipeline is bootstrapped from detectors or MLLMs, which improves scalability but is not equivalent to fully human annotation. Second, in single-view settings, the duplicate-image fallback is a practical implementation choice and does not provide a clean test of cross-view fusion. Third, while our current results on CityBench suggest transfer beyond the U.S. training distribution, the dataset and main evaluation remain predominantly U.S.-centric, and a broader out-of-distribution study would further strengthen the claims.

---

> > ### Author Rebuttal · Reviewer_zMBQ · 2026-04-03
> >
> > Thank you for the detailed response. The authors have made a reasonable effort to address my concerns, and I appreciate the additional provenance table and the supplementary regression metrics. However, I would like to note that several concerns remain only partially resolved.
> >
> > Regarding Q1, while the provenance table is a helpful addition, the evaluation targets for the street-view object reasoning task are still partially derived from detector outputs. The implications of this for the validity of reported gains are not fully discussed.
> >
> > Regarding Q2, the authors clarify that geographic coordinates are not provided as direct inputs during geo-localization evaluation, and point to CityBench results as evidence of generalization. However, the core concern is not simply about input leakage: since the model was pre-trained on paired image-coordinate data, the visual features themselves may have implicitly encoded geographic location information. The rebuttal does not sufficiently rule out this possibility. Furthermore, while the geo-localization task is framed as a multiple-choice problem, this does not fully eliminate the concern — implicit geographic biases encoded in the visual features could still influence the model's preference toward certain answer choices, even when no coordinates are explicitly provided at inference time. The distinction between genuine visual geo-localization and implicit coordinate memorization therefore remains unclear.
> >
> > Regarding Q3, the authors frame the cross-view perceiver and interleaved pre-training as a jointly designed system. However, the ablation results — where the 'w/o pre-training' variant matches the fine-tuned VILA1.5-8B baseline — suggest that the independent contribution of the pre-training stage is limited, at least for single-view tasks. Attributing gains to the joint design does not fully resolve the question of whether the pre-training stage provides meaningful benefit beyond the perceiver module alone.
> >
> > In light of the above, I maintain my original score.

---

> > > ### Author Response · Authors · 2026-04-04
> > >
> > > We thank the reviewer for the careful follow-up.
> > >
> > > (1) Provenance and reliability of the street-view OR benchmark.
> > > To assess reliability, we randomly sampled 100 test examples from the SFT dataset for manual inspection and observed approximately 90% agreement. In addition, three expert annotators participated in quality control of the test set, and samples with clearly unacceptable label quality were removed. We will revise the paper to make this provenance more explicit and to qualify the OR results accordingly.
> > >
> > > (2) Geo-localization: direct leakage versus learned geographic priors.
> > > We would like to distinguish direct train-test leakage from learned geographic priors. The test images and their associated locations are fully disjoint from both the pre-training and fine-tuning data, which rules out the simpler explanation that the model solves GL by memorizing previously seen test images or coordinates. This addresses the concern of direct overlap between training and evaluation instances.
> > >
> > > At the same time, we agree that incorporating coordinates during pre-training is intended to help the model learn visual-geographic associations. In that sense, GL should not be interpreted as a purely appearance-only geo-localization test. To further examine whether GL performance is driven primarily by coordinate memorization, we conducted an additional ablation in which coordinates and related location text were completely removed from the pre-training corpus. Relative to the full model (SI-GL: 0.871; SV-GL: 0.879), performance decreases by a limited margin under this setting (SI-GL: 0.857; SV-GL: 0.863). We interpret this result as evidence that coordinate-aware pre-training is beneficial, but not the sole driver of GL performance: removing location text causes a measurable drop, yet the model retains strong geo-localization ability, indicating that visual cues remain an important component of the learned representation. We will revise the paper accordingly and describe GL more precisely as evaluating visual-geographic grounding, which may combine visual evidence with learned geographic regularities, rather than purely appearance-only geo-localization.
> > >
> > > (3) Contribution of interleaved pre-training.
> > >
> > > Our ablation suggests a more nuanced picture. Specifically, the comparison between the variant without pre-training and the variant without the cross-view perceiver does not show a consistent standalone gain from interleaved pre-training alone. We therefore do not claim that pre-training, by itself and without the perceiver, yields uniform improvements across tasks.
> > >
> > > Instead, our interpretation is narrower: the main role of interleaved pre-training is to provide a useful cross-view learning signal whose benefit becomes most evident when coupled with the cross-view perceiver. This is reflected in the full model, where the clearest improvement appears on SRR, a task that directly requires cross-view spatial reasoning. Relative to the variant without pre-training, the full model improves SRR from 0.321 to 0.345; relative to the variant without the cross-view perceiver, SRR improves from 0.319 to 0.345. We will revise the paper accordingly and avoid overstating the independent effect of interleaved pre-training. The more precise conclusion supported by the current evidence is that interleaved pre-training is most effective as part of the full cross-view learning design, rather than as a uniformly beneficial standalone component. Please refer to Q5/W2 for Reviewer Pwy9.

---

### Official Review · Reviewer_KqUF · 2026-03-13

**Soundness:** 3
**Presentation:** 3
**Significance:** 3
**Originality:** 3
**Overall Recommendation:** 4
**Confidence:** 2

**Summary:**

The authors seek to outline a pressing issue in the domain of urban visual understanding: existing Multimodal Large Language Models (MLLMs) predominantly focus on macroscopic remote sensing (satellite) data, thereby ignoring the fine-grained, ground-level details provided by street-view imagery. To bridge this gap, the paper proposes UrbanMLLM, a unified framework that jointly learns from both satellite and street-view images. The authors explore an important aspect of multimodal architecture by introducing a "cross-view perceiver" module, which explicitly exchanges information between region-level satellite contexts and appearance-level street-view details via cross-attention. Additionally, the authors construct a massive, paired cross-view dataset (4.5 million image pairs) and 700k instruction-tuning QA pairs, leveraging a novel interleaved pre-training paradigm. Evaluated on a comprehensive benchmark of 13 diverse tasks, UrbanMLLM demonstrates state-of-the-art performance against both specialized and general-purpose MLLMs.

**Compliance With Llm Reviewing Policy:**

Affirmed.

**Key Questions For Authors:**

1. Given that the cross-view perceiver slightly degraded performance on street-view indicator predictions, have you considered implementing a dynamic routing or dynamic gating mechanism that allows the model to completely ignore the secondary view if it is deemed unhelpful for the specific prompt?
2. How do you anticipate UrbanMLLM performing on urban environments outside the United States, particularly in developing nations with unstructured urban planning? Are there plans to evaluate the model on non-US datasets?
3. What is the exact computational overhead (in terms of FLOPs or inference latency) introduced by the cross-view perceiver when processing multiple high-resolution street-view images alongside a satellite image, compared to a standard MLLM?
4. Satellite and street-view images are often captured years apart. How does the interleaved pre-training handle temporal discrepancies between the two views (e.g., a new building appears in the street view but not in the older satellite map)?

**Limitations:**

Yes

**Strengths And Weaknesses:**

## Strengths

1. The integration of paired satellite and street-view imagery into a single MLLM framework is highly innovative and addresses a clear gap in comprehensive urban environment modeling.
2. The proposed cross-view perceiver is a well-designed, intuitive mechanism to break the "visual isolation" typical of standard MLLMs, allowing the model to mutually enhance features from different viewpoints before feeding them into the LLM backbone.
3. The construction of a large-scale, nationwide (US) interleaved image-text dataset using a rigorous Human-AI collaborative pipeline is a significant contribution to the open-source community.
4. The paper establishes a robust benchmark covering 13 tasks (perception, reasoning, and prediction) across single and cross-view settings, proving the model's versatility. It also demonstrates strong zero-shot generalization on external benchmarks like Citybench.

## Weaknesses
1. The dataset is exclusively collected from the United States (based on US census tracts and Google Maps). Urban layouts, architectural styles, and socio-economic indicators vary drastically across the globe, limiting the model's immediate generalization to international contexts.
2. The ablation study (Table 6) reveals that removing the cross-view perceiver actually *improves* performance on certain street-view indicator prediction tasks (e.g., Beautiful/Wealthy level). This suggests that forcing cross-view fusion can sometimes introduce macroscopic noise into tasks that rely purely on intrinsic street-level aesthetics.
3. The data collection relies heavily on Google Maps API and GPT-4o for generation/filtering. This raises concerns about the reproducibility of the pipeline and potential inherited biases from closed-source models.
4. While the paper mentions duplicating single-view images to bypass the cross-view perceiver when paired data is missing, it lacks a deep analysis of how the model performs in real-world scenarios where street-view data is sparse or severely misaligned with satellite data.

---

> ### Author Rebuttal · Authors · 2026-03-31
>
> Thank you for the positive and insightful review
>
> ### W1/Q2: Generalization Beyond the U.S.
>
> While the primary evaluation focuses on U.S. data, the paper already includes results on **CityBench**, which spans **13 cities worldwide** (Fig. 4(b)). Our model consistently outperforms both leading closed-source and open-source baselines on this benchmark. These results indicate that the model maintains stable performance beyond the training distribution. We will revise the manuscript to highlight this evidence more clearly and moderate the scope of our claims accordingly. While CityBench already provides an initial non-U.S. evaluation, we agree that broader assessment on more diverse urban forms, especially in developing regions, would further strengthen the claim and is a natural next step.
>
> ### W2: Task-Dependent Effect of the Cross-View Perceiver
>
> The observed degradation is better understood as **task-dependent behavior** rather than a general limitation. As shown in Table 6, tasks relying primarily on **intrinsic street-level appearance** (e.g., aesthetics or some socioeconomic indicators) may be negatively influenced by additional satellite context. In contrast, tasks involving **spatial reasoning** or **cross-view grounding** consistently benefit. We will clarify this distinction and avoid implying uniformly positive effects of cross-view fusion, while noting that **adaptive routing** is a natural extension.
>
> ### W3: Reproducibility and External Dependencies
>
> The pipeline is structured and reproducible in principle, including **deterministic geo-sampling**, **explicit view pairing**, **fixed prompt templates**, and **well-defined filtering rules**. Street-view data are indexed via **pano IDs**, enabling retrieval of the same views. We will release detailed instructions and code for data reconstruction, and explicitly document external dependencies to facilitate reproducibility.
>
> ### W4: Robustness to Sparse or Misaligned Views
>
> We further analyze the impact of the number of street-view images (Table 11). Performance improves with additional views and saturates around **4-5 images**, suggesting that sufficient information can be extracted from a limited set. Under sparse-view settings, the model can still leverage satellite context to produce reasonable predictions, indicating robustness. We will also clarify that the current study mainly evaluates robustness to **limited-view availability**; a more systematic analysis of **severe cross-view spatial or temporal misalignment** remains future work.
>
> ### Q1: Dynamic Routing / Gating
>
> We agree that **dynamic routing** is a promising direction. Current results already reveal when additional views are beneficial or detrimental; adaptive gating could further refine this behavior, but is not necessary to validate the effectiveness of the current design.
>
> ### Q3: Computational Overhead
>
> Compared to a standard MLLM projector, the proposed **cross-view perceiver** introduces only **0.68 ms** additional latency under our setting (**1 satellite + 4 street views; 729 tokens/image; mm_hidden_size=1152; projector_hidden_size=4096; FP16 on CUDA**). The projector path increases from **0.43 ms** to **1.11 ms** (**2.60x** at module level), corresponding to about **24.25 GFLOPs**, which remains modest in the context of the full model.
>
> ### Q4: Temporal Misalignment
>
> Temporal inconsistencies are inherent in large-scale multimodal data and can encourage the model to rely on **stable semantic cues** rather than exact timestamp alignment. Empirically, the model demonstrates robustness under such conditions, suggesting that strict temporal synchronization is not required for effective learning.

---

> > ### Author Rebuttal · Reviewer_KqUF · 2026-04-04
> >
> > Thanks for the response. All my concerns are resolved. Please include these additional discussion and experiments in you revision.

---

### Official Review · Reviewer_PWy9 · 2026-03-13

**Soundness:** 3
**Presentation:** 2
**Significance:** 3
**Originality:** 2
**Overall Recommendation:** 3
**Confidence:** 4

**Summary:**

This paper studies urban multimodal large language models that jointly use satellite imagery and street-view imagery for urban understanding. The authors argue that existing urban MLLMs mainly rely on satellite views and therefore miss fine-grained ground-level semantics as well as the complementary relationship between overhead and street-level observations. To address this, the paper introduces UrbanMLLM, built on top of a VILA-style MLLM backbone with two main additions: (1) a cross-view perceiver that exchanges information between paired satellite and street-view visual representations through cross-attention and gated fusion, and (2) an interleaved pre-training paradigm that organizes cross-view images and associated text into urban documents to encourage implicit cross-view knowledge fusion. Evaluation is conducted on a benchmark of 13 urban understanding tasks. The reported results show that UrbanMLLM generally improves over several open-source and proprietary MLLM baselines.

**Compliance With Llm Reviewing Policy:**

Affirmed.

**Final Justification:**

Thank you to the authors for the additional experiments and clarifications during the rebuttal. I appreciate the effort to include stronger frontier baselines under the limited time window. However, I still feel the paper would benefit from more complete experiments, further analysis of some counterintuitive results, and a more substantial revision of the writing and presentation. Overall, the current version is not yet strong enough for acceptance, so I will maintain my score.

**Key Questions For Authors:**

1. See weaknesses.
2. Can you provide stronger evidence that the interleaved pre-training corpus is responsible for cross-view reasoning improvements beyond simple scale effects?

**Limitations:**

yes

**Strengths And Weaknesses:**

**Strengths**

1. The paper studies an important and timely problem. This is a well-motivated setting, since urban environments are inherently cross-view and existing urban MLLMs tend to rely more heavily on overhead imagery.
2. The overall method is reasonable and coherent. The proposed cross-view perceiver is a natural mechanism for exchanging information between complementary views, and the interleaved pre-training paradigm is well aligned with recent multimodal pre-training practice.
3. The dataset contribution appears valuable and could benefit the broader community.

**Weaknesses**

1. The data construction pipeline is insufficiently specified, especially for the SFT stage. While the paper mentions a human–AI collaborative process with rule-based templates and LLM-generated QA pairs, it does not provide enough detail on data composition, task distribution, prompting strategy, human revision rate, or quality control.
2. The data quality validation is not fully convincing. The paper mainly uses CLIP score changes before and after caption cleaning and the near-unchanged CLIP score before and after cleaning may indicate either high initial quality or limited sensitivity of the metric to semantically important caption errors. Stronger human evaluation or task-specific validation would make this claim more convincing.
3. The baseline comparison is reasonably broad, but it does not include some stronger and more recent frontier multimodal models such as gemini 3 pro, which makes it harder to judge how competitive the method is relative to the current state of the art.
4. The repository link is expired, which further weakens reproducibility.

---

> ### Author Rebuttal · Authors · 2026-03-31
>
> ### W1. SFT / data pipeline clarity
>
> Thank you for the suggestion. More details about the data could be found in the appendix.
>
> Our SFT data is not produced by unconstrained LLM-based QA generation, but by a structured pipeline with three parts: **street-view-only, satellite-only, and cross-view data**. These cover scene classification, object reasoning, spatial relation reasoning, geo-localization, landmark recognition, and socioeconomic prediction. Supervision mainly comes from **grounded sources** such as geospatial metadata, public datasets, detector outputs, and statistical indicators. LLMs are used mainly for **question formatting, distractor generation, and linguistic diversification**, rather than as the main source of supervision. Please refer to Q1/W1 for Reviewer zMBQ.
>
> The prompting strategy is largely **template-based and rule-constrained**, with standardized outputs such as multiple-choice answers, object counts/localization, or single-value predictions in the form `X.X`. We also apply **automatic validation and human-AI checking** to enforce label and format consistency. As discussed in **W2**, the mismatch rate is very low, supporting the overall quality of the SFT data. We will make these points explicit in the revision.
>
> ### W2. Data quality validation
>
> We clarify that **CLIP score is only an auxiliary metric**, not the main evidence for caption quality. Thus, the small change before and after cleaning should not be viewed in isolation. Since the cleaned captions are used for training, the limited CLIP improvement is more consistent with **high initial caption quality** and also reflects CLIP's limited sensitivity to some semantic errors.
>
> Our main evidence comes from a **multi-stage validation pipeline**. Each image-caption pair is checked by **VILA-1.5-40B** and **LLaVA-Next-34B**; flagged cases are then reviewed by **GPT-4o**; remaining disagreements are resolved by a **graduate-level human annotator**. Under this process, only about **1.3 pairs per 1,000 samples(0.13%)** are ultimately identified as mismatched, providing direct evidence of strong image-caption consistency.
>
> We also conducted **human inspection on 100 sampled cases**. In all 100 cases, the generated caption covered the **top-3 salient objects** identified by the annotator, supporting descriptive usefulness beyond CLIP. This validation evidence also addresses the data quality concern in **W1**.
>
> ### W3. Stronger baselines
>
> We agree that stronger frontier MLLMs would make the evaluation more complete. We therefore add results for two stronger high-capacity models under the same setting:
>
> |Model|SC(Single)|SC(Multi)|GL|IP(Pop)|IP(Nightlight)|SV-LR|SV-GL|
> |---|---:|---:|---:|---:|---:|---:|---:|
> |Qwen3-VL-32B|0.529|0.314|0.458|0.617|0.302|0.723|0.652|
> |Gemini-3.1-Pro|0.649|0.565|0.867|0.451|0.562|0.781|0.743|
>
> These results provide a stronger reference to current high-capacity multimodal systems. More broadly, our goal is not only to compare a few isolated tasks, but to evaluate whether a model designed for **cross-view urban understanding** remains effective across a heterogeneous urban benchmark. We will clarify this evaluation scope and the baseline selection criterion in the revision.
>
> ### W4. Repository link / presentation
>
> The repository link has been corrected. We will also re-check all released links and materials in the revision to ensure reproducibility.
>
> ### Q2. Contribution of interleaved pre-training
>
> The ablation does not support the view that the gain is merely from scale. Instead, it points to a more specific effect on **cross-view reasoning**.
>
> |Variants|Med.income|Pov.ratio|Population|SRR|
> |---|---:|---:|---:|---:|
> |UrbanMLLM-8B|**0.761**|**0.668**|**0.543**|**0.345**|
> |w/o cross-view perceiver|0.727|0.646|0.530|0.319|
> |w/o pre-training|0.740|0.660|0.531|0.321|
>
> Relative to **w/o pre-training**, the full model shows only modest gains on socioeconomic prediction (`0.740->0.761`,`0.660->0.668`,`0.531->0.543`), but a clearer improvement on **SRR** (`0.321->0.345`). If the effect were mainly due to scale, a more uniform gain across tasks would be expected. Instead, the largest gain appears on **SRR**, which directly measures **spatial relation reasoning across views**.
>
> The comparison with **w/o cross-view perceiver** supports the same conclusion: removing either the cross-view module or the interleaved pre-training signal reduces performance, with the clearest drop again on SRR. Together, these results suggest that the gain comes from the interaction between the **cross-view architecture** and the **interleaved pre-training corpus**, rather than from scale alone. This is also consistent with the CityBench results, where the advantage of the full model is more evident on tasks requiring cross-view understanding.

---

> > ### Author Rebuttal · Reviewer_PWy9 · 2026-04-02
> >
> > Thanks for the rebuttal. It partially addresses my concerns by clarifying that the SFT supervision is grounded and by adding stronger baselines, but the SFT construction process is still not specified in enough concrete detail (e.g., prompt templates). I also remain uncertain about the stronger-baseline comparison, since the rebuttal does not report results on the other street-view imagery-based tasks or the cross-view tasks, and the fact that Gemini-3.1-Pro appears substantially worse than GPT-4o on the reported tasks is somewhat surprising and would benefit from further explanation. And the presentation is somewhat fragmented, making the paper less easy to follow and some parts harder to understand than necessary.

---

> > > ### Author Response · Authors · 2026-04-04
> > >
> > > Thank you for the follow-up. We understand that your remaining concerns mainly center on three points: the SFT construction process is still not concrete enough, the stronger-baseline comparison is still limited in scope, and the presentation remains somewhat fragmented. We agree that these issues should be addressed more clearly in the revision, and respond point by point below.
> > >
> > > ### 1. On the specificity of the SFT construction process
> > >
> > > We agree that the current presentation of the SFT pipeline in the main paper is still not sufficiently direct and consolidated, although the underlying materials are already available. Concretely, we have added an `sft_data` folder in the GitHub repository that contains the dataset construction scripts for different tasks, and `prompt.py` summarizes the main prompts used across task families. In addition, Appendix Figures 7-15 already provide representative prompt-and-answer examples. In the revision, we will align these existing materials more explicitly with the main text, and clarify more clearly which parts of the prompts are used for linguistic diversification and which parts are tied to structured supervision, so that the SFT pipeline is easier to follow and inspect at the task-family level.
> > >
> > > ### 2. On the stronger baselines and the interpretation of Gemini-3.1-Pro results
> > >
> > > We note that, during the rebuttal period, the evaluation of proprietary frontier models was constrained by the rebuttal time window and API inference speed, so we were not able to complete all tasks at that stage and only reported a representative subset first. To address this, we subsequently added results on multiple satellite-only, street-view-only, and cross-view tasks, shown below.
> > >
> > > #### Satellite-only tasks
> > >
> > > | Model | SC (Single) | SC (Multi) | OR | SRR | GL | IP (Pop) | IP (Nightlight) |
> > > |------|-------------|------------|----|-----|----|----------|-----------------|
> > > | Qwen3-VL-32B | 0.529 | 0.314 | 0.401 | 0.582 | 0.458 | 0.617 | 0.302 |
> > > | Gemini-3.1-Pro | 0.649 | 0.565 | 0.725 | 0.591 | 0.867 | 0.451 | 0.562 |
> > >
> > > #### Street-view-only tasks
> > >
> > > | Model | SC | OR | LR | SRR | GL | IP (BF) | IP (WE) |
> > > |------|----|----|----|-----|----|---------|---------|
> > > | Qwen3-VL-32B | 0.499 | 0.633 | 0.723 | 0.725 | 0.652 | 0.819 | 0.705 |
> > > | Gemini-3.1-Pro | 0.670 | 0.613 | 0.781 | 0.777 | 0.743 | 0.845 | 0.767 |
> > >
> > > #### Cross-view tasks
> > >
> > > | Model | Med. Income | Pov. Ratio | Population | SPR |
> > > |------|-------------|------------|------------|-------|
> > > | Qwen3-VL-32B | 0.598 | 0.566 | 0.493 | 0.250 |
> > > | Gemini-3.1-Pro | 0.772 | 0.626 | 0.485 | 0.332 |
> > >
> > > Based on this more complete comparison, we do not think Gemini-3.1-Pro shows an overall trend of being clearly weaker than GPT-4o; rather, the impression from the previous rebuttal was mainly caused by limited result coverage rather than the overall pattern on the full benchmark. At the same time, these extended results also do not support a simple explanation for the remaining local gaps. For example, within cross-view socioeconomic prediction, Gemini-3.1-Pro is better than GPT-4o on `Med. Income`, while differences remain on `Pov. Ratio` and `Population`, indicating that its strengths and weaknesses do not align consistently with any single task category. Therefore, in the revision, we will present the stronger-baseline coverage more completely and frame these results as supplementary references for frontier multimodal systems, rather than as an overall ranking of proprietary models.
> > >
> > > ### 3. On the fragmented presentation
> > >
> > > Because the paper covers a diverse set of tasks, data sources, and processing pipelines, the current presentation may indeed feel somewhat fragmented. Figure 1 in the main paper already provides the overall pipeline, but we agree that the correspondence between the main text and the task-specific data workflows still needs to be made clearer. In the revision, we will improve the organization and indexing of task-specific data workflow descriptions in the appendix, and further streamline the presentation order of model design, training stages, and data construction in the main text to improve readability and ease of navigation.
> > >
> > > Thank you again for the constructive feedback. We will incorporate all the above clarifications into the revised manuscript. We hope these responses address your remaining concerns and look forward to your final recommendation.

---

### Decision · Program_Chairs · 2026-04-30

**Decision:**

Accept (regular)

**Comment:**

This submission received three 'Weak Accept' recommendations and one 'Weak Reject.' The authors' rebuttal successfully addressed several of the reviewers' concerns. Upon closer inspection of the negative review, the primary reservation centers on the need for more comprehensive experiments. Despite this, I believe the current work provides a meaningful contribution to the community and warrants acceptance.